# Localization control born of intertwined quasiperiodicity and non-Hermiticity

**Junmo Jeon and SungBin Lee**⋆

Korea Advanced Institute of Science and Technology, Daejeon 34141, South Korea

⋆ sungbin@kaist.ac.kr

## Abstract

Quasiperiodic systems are neither randomly disordered nor translationally invariant in the absence of periodic length scales. Based on their incommensurate order, novel physical properties such as critical states and self-similar wavefunctions have been actively discussed. However, in open systems generally described by the non-Hermitian Hamiltonians, it is hardly known how such quasiperiodic order would lead to new phenomena. In this work, we show that the intertwined quasiperiodicity and non-Hermiticity can give rise to striking effects: perfect delocalization of the critical and localized states to the extended states. In particular, we explore the wave function localization character in the Aubry-André-Fibonacci (AAF) model where non-reciprocal hopping phases are present. Here, the AAF model continuously interpolates the two different limits between metal to insulator transition and the critical states, and the non-Hermiticity is encoded in the hopping phase factors. Surprisingly, their interplay results in the perfect delocalization of the states, which is never allowed in quasiperiodic systems with Hermiticity. By quantifying the localization via the inverse participation ratio and the fractal dimension, we discuss that the non-Hermitian hopping phase leads to delicate control of localization characteristics of the wave function. Our work offers (1) emergent delocalization transition in quasiperiodic systems via non-Hermitian hopping phase and (2) detailed localization control of the critical states. In addition, we suggest an experimental realization of controllable localized, critical and delocalized states, using photonic crystals.

 Check for updates

## 1  Introduction

Quasiperiodic order, which is a novel spatial pattern without any periodic unit length scales, has attracted interest from a wide range of physics disciplines [1–5]. In a quasiperiodic system, not only the diffraction pattern [3,6], but also the electromagnetic and topological properties would be different from those of conventional periodic crystals [7–12]. This is mainly due to the localized nature of their quantum states, which decay along a power-law scale and thus are neither localized nor extended, so-called critical states [13–21]. The critical states arise from the incommensurate self-similar quasiperiodic ordered structure [16,17]. Theoretically, the quasiperiodic systems and possible critical states in one-dimensional chains have been actively studied in terms of the Aubry-André model [20,22] and the Fibonacci quasicrystal [14,16]. The critical states that arise in these systems lead to the stable fractal magnon transmittance which has advanced the field of magnonics [23–25]. In recent years, it has become possible to artificially create quasicrystalline structures in the laboratory, such as metamaterials and photonic crystals [26–31], and the potential for experimental applications using quasiperiodicity and critical states has been increased. However, how the quasiperiodic orders and their critical states behave in the open systems is still poorly understood.

To understand open systems where energy is not conserved, the effective non-Hermitian theory has been used as a theoretical framework by neglecting quantum jumps from the standard Lindblad equation [32–34]. In the last decade, there has been a growing interest in the systems described by non-Hermitian Hamiltonians. For example, non-Hermitian Hamiltonians have been widely used in exciton-polariton theory [35–37], photonics [38], and other optical systems [39]. More recently, researchers have developed non-Hermitian descriptions of magnonic systems [40,41]. These have been used to describe magnons in driven and dissipative spintronic systems [41]. There is also a non-Hermitian tight-binding model with the non-reciprocal hopping magnitudes that exhibits the non-Hermitian skin effect, the massive condensation of bulk modes to the edge under the open boundary condition due to the nonzero spectral area under the periodic boundary condition [32, 42–55]. Such non-Hermitian systems have been realized experimentally using photonic crystals, fiber optics, or electrical circuits [38, 56–60]. Furthermore, the effect of non-Hermiticity in quasiperiodic systems has also been spotlighted in terms of the parity-time symmetry and the topological phase transitions [44, 61–66]. Nonetheless, it remains to be understood how such non-Hermiticity can be used to manipulate drastic changes of the quantum states in quasiperiodic systems.

In the current work, we study the system where both quasiperiodic order and non-Hermiticity play an important role and discuss the striking result due to their interplay. In contrast to the traditional non-Hermitian models explained by the Hatano-Nelson argument [47], here the non-Hermiticity is taken into account as the non-reciprocal phase of the hopping parameters, instead of the hopping magnitudes. As a representative example of the quasiperiodic order, we study the Aubry-André-Fibonacci model [67], but note that our argument is generally applicable to other types of quasiperiodic systems. We show that the interference

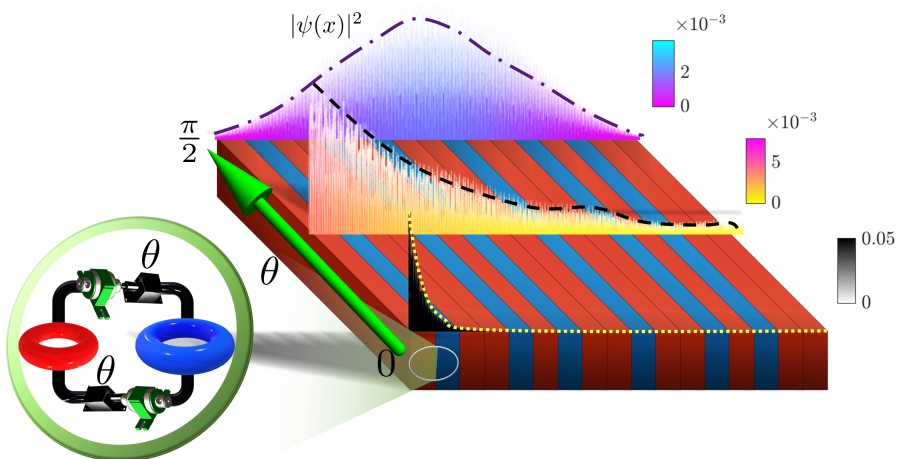

Figure 1: A proposal for experimental control of the localization characteristics in a non-Hermitian optical system with ring resonators. A simplified representation of the ring resonators that make up the quasiperiodic pattern is shown in the sequence of red and blue blocks. Two or more ring resonators with different resonant frequencies are arranged in a quasiperiodic pattern to form a photonic crystal. Here, the ring resonators with two different resonance frequencies are drawn as red and blue rings in the zoomed inset. Adjacent ring resonators are connected by two optical cables with optical isolator and phase shifter, shown as green and black components, respectively, in the zoomed inset. Each optical isolator allows the transmission of light only in the left and right directions, and the phase shifter adds the same phase to the transmitted wave. Thus, we have an effective non-reciprocal hopping phase for the light. By arranging the ring resonators as a Fibonacci quasicrystal and manipulating the non-reciprocal hopping phase, $\theta$, accumulated by the phase shifters, we can change the localization characteristics of the wave function, $\psi(x)$, from exponentially localized (black) to extended (blue), exploring critical states (yellow) in terms of $\theta$. Two different decay behaviors of the localized and critical states are shown by the dotted and dashed lines, respectively, indicating the envelopes of the probability distributions. One could measure the optical conductivity to read off the change in light localization properties.

arising from the non-reciprocal hopping phases and the exceptional coalescence of the states controls the localization of the wave function. Surprisingly, intertwined quasiperiodicity and non-Hermiticity gives rise to a perfect delocalization without fractality of the wave function, which never occurs in a quasiperiodic system with Hermiticity. Not only the emergence of delocalization, but also their interplay leads to the control of critical states, implying potential applications.

Fig. 1 illustrates our main results with sketch of potential experimental implications. Here, as an example of the quasiperiodic system, the two different ring resonators (red and blue blocks) are arranged in Fibonacci quasicrystalline patterns. Each ring resonator is connected by the optical isolators [68] and phase shifters which in principle controls the non-reciprocal hopping phase, $\theta$ (see the inset of Fig. 1). By changing the non-Hermiticity given by the non-reciprocal hopping phase, $\theta$ from 0 to $\pi/2$, we suggest that one can explore general localization characteristics of the wave functions from localized (black) to critical (yellow) and extended (blue), as shown on the top of the quasiperiodic arrays. Therefore, the non-Hermitian quasiperiodic system can be used to manipulate quantum transport experiments.

The rest of the article is organized as follows. In Sec.2, we describe our non-Hermitian tight-binding Hamiltonian with non-reciprocal hopping phase. In Sec.3, using the Aubry-André-Fibonacci model as an example, we analyze the change in the localization strength of the state given by the inverse participation ratio with respect to the non-reciprocal hopping phase. We show that the interference effect arising from the interplay of quasiperiodic potential and the non-reciprocal hopping phase gives rise to the change in the localization strength of the states. We also show how the states become delocalized in the Fibonacci chain as a function of the non-reciprocal hopping phase. We explain that the exceptional hybridization of the eigenstates leads to the change of the localization characteristics of the states. In Sec.4, we summarize our work.

## 2 Non-Hermitian Hamiltonian with non-reciprocal hopping phase

Let us consider the tight-binding Hamiltonian with $N$ sites

$$H = H_V + H_T, \tag{1}$$

$$H_V = \sum_{i=1}^{N} V_i |i\rangle \langle i|,$$

$$H_T = t \sum_{i=1}^{N-1} (|i\rangle \langle i+1| + |i+1\rangle \langle i|),$$

$$t = T e^{i\theta},$$

where $V_i$ are non-uniform real-valued local potentials. $t$ is the uniform complex-valued hopping parameter, respectively. $|i\rangle$ represents the particle placed on the $i$-th site. The hopping parameter is uniform complex-valued, $t = Te^{i\theta}$ where $T$ and $\theta$ are positive reals. The strength of the non-Hermiticity is given by $T\sin\theta$, which is maximized when $\theta = \pi/2$. Note that the strength of the non-Hermiticity is depending on both the hopping magnitude and the non-reciprocal hopping phase. We will show that the non-Hermiticity encoded in the non-reciprocal hopping phase changes the localization properties of the states compared to the Hermitian counterparts given by $\theta = 0, \pi$.

One of the most important quantities used in the non-Hermitian systems is the phase rigidity, which is defined by

$$r(\psi_k) = \left| \langle \psi_k^{(L)} | \psi_k^{(R)} \rangle \right|. \tag{2}$$

Here, the superscripts $L$ and $R$ stand for the left and right eigenstates of the non-Hermitian Hamiltonian, and the subscript $k$ is the index of the eigenstate. Unlike the Hermitian systems where the phase rigidity is always 1, it could be less than one, and even vanish in the non-Hermitian systems. Particularly, when two different eigenstates coalesce, the phase rigidity becomes zero [69, 70]. This unique characteristics of the non-Hermitian system is called an exceptional point. Thus, one can use the phase rigidity to quantify the coalescence of the states in the non-Hermitian system.

We quantify the localization strength of the state by using the inverse participation ratio (IPR), which is defined for a normalized state $\psi$ as

$$\text{IPR}(\psi) = \sum_i |\psi(i)|^4. \tag{3}$$

Note that the amount of localization for the wave function, $\psi$, can be quantified by the IPR [44, 71, 72]. In the spectrum, the maximum (minimum) value of the IPR indicates the maximally (minimally) localized state. Let us refer to these states in the spectrum as maximally localized and maximally extended states, respectively. Also, the average localization strength for entire states in the spectrum is given by the mean IPR (MIPR), defined by

$$\text{MIPR} = \frac{1}{N} \sum_{k=1}^{N} \text{IPR}(\psi_k), \tag{4}$$

where $\psi_k$ is the $k$-th eigenstate. The delocalization (localization) can be captured by the reduction (enhancement) of MIPR [44].

## 3 Delocalization in the non-Hermitian quasiperiodic chains

Given a finite hopping magnitude, $T$, we can change the localization characteristics of the states as $\theta$ approaches $\pi/2$, where the strength of non-Hermiticity becomes maximum. To understand how the non-Hermitian hopping phase factor allows to change the localization of the states, let us consider the return probability, which measures the probability that the particle placed at the $i$-th site will return to the $i$-th site. A smaller return probability indicates that the state is more delocalized because the wave function is scattered. Using the path integral idea, the return probability is given by the sum of the transition amplitudes of all possible paths whose start and end points are the same as the $i$-th site. If the hopping phase is non-reciprocal, there is destructive interference between the transition amplitudes. This interference originated from the non-reciprocal hopping phase essentially gives rise to the delocalization of the state by reducing the return probability. On the other hand, the non-reciprocal phase can also lead to the constructive interference with respect to $\theta$ for some states. In this case, localization is further enhanced due to the interference from the non-reciprocal hopping phase. Thus, the non-reciprocal hopping phase gives rise to the *state-dependent* control of the localization properties. Moreover, in the non-Hermitian systems, such interference effect leads to the unconventional coalescence of the states which have different localization characteristics, the so-called exceptional points. Hence, before and after this exceptional point, the localization characteristics would be changed, as we will show.

To illustrate the *state-dependent* control of the localization strength with a concrete argument, we consider a toy model consisting of two different atoms, $A$ and $B$, arranged in an alternating way, i.e. $ABABAB\cdots$. This can be understood as the 1/1-approximant of the Fibonacci chain, which we discuss in Sec.3.2 [21]. With periodic boundary conditions, the Hamiltonian in momentum space is given by the $2 \times 2$ matrix below.

$$H(k) = \begin{pmatrix} V_A & t(1+e^{-ika}) \\ t(1+e^{ika}) & V_B \end{pmatrix}, \tag{5}$$

where $a$ is the distance between the atoms, $k$ is the momentum, and $V_A$ and $V_B$ are the local potentials of the $A$ and $B$ atoms, respectively. The eigenvectors are given by,

$$|v_{\pm}(k)\rangle = \frac{1}{\mathcal{N}_{\pm}} \begin{pmatrix} 2t(1+e^{-ika}) \\ \Delta V \pm \sqrt{\Delta V^2 + 16t^2\cos^2 ka/2} \end{pmatrix}, \tag{6}$$

where $\Delta V = V_B - V_A$ which is assumed to be positive without loss of generality. $\mathcal{N}_{\pm}$ is a normalization constant. In the eigenstates, $\Delta V^2 + 16t^2\cos^2 ka/2$ indicates the interference between the potential difference and hopping contributions. Note that this interference effect is the result of the local potential gradient, $\Delta V$ and the non-reciprocal hopping phase $\theta$

in $t = Te^{i\theta}$. Furthermore, the relative probability amplitude at $B$ sublattices is reduced (increased) as a function of $\theta$ for $v_{+(-)}(k)$. This is because the return probability for the $B$ atoms of $v_{+(-)}(k)$ is reduced (enhanced) due to the state-dependent phase difference between $\Delta V$ and $\sqrt{\Delta V^2 + 16t^2 \cos^2 ka/2}$. This is a typical example of the state-dependent control of localization characteristics as a result of the interplay between the non-flat potential distribution and the non-Hermiticity.

The interference effect between the non-uniform potential and the non-reciprocal hopping phases, can be understood with the effective potential distribution, $V_i^{\text{eff}}$. Here, $V_i^{\text{eff}}$ is defined as the deformed on-site potential which leads to the same probability distribution of the given state assuming $\theta = 0$. In general, the effective potential is state-dependent, distinguishing different momentum $k$ from Eq.(6). Surprisingly, however, when the hopping magnitude $T$ is larger than $\frac{\Delta V}{4\cos(Ka/2)}$ for a given momentum $K$, the probability amplitude at the $B$ sublattices of $v_{\pm}(k \leq K)$ and their localization strengths are saturated at $\theta = \pi/2$ as the same value regardless of the sign difference in Eq.(6) and momentum $k \leq K$. Thus, the effective potential becomes uniformly periodic for each $v_{\pm}(k \leq K)$. In particular, the probability distribution for $k \leq K$ becomes uniform in this case, and thus indistinguishable from a uniformly periodic chain with no potential gradient. This shows that if the finite hopping magnitude is sufficiently large compared to the potential difference, the effect of the potential gradient would be washed out by the interference effect arising from the non-reciprocal hopping phase. In other words, the non-reciprocal hopping phase deforms $V_i$ to the uniform effective potential $V_i^{\text{eff}}$ such that $\Delta V^{\text{eff}} = 0$. Such a delocalization of the states would be achieved delicately for each momentum $k$ by increasing $T$. Thus, a non-reciprocal hopping phase provides a high controllability of the localization strength for each state.

Note that when $T = \frac{\Delta V}{4\cos(Ka/2)}$, $v_+(K)$ coalesces into $v_-(K)$, and hence the phase rigidity defined in Eq.(2) for $v_{\pm}(K)$ becomes zero. Thus, the unconventional coalescence of the states due to the non-Hermiticity occurs when the state is uniformly delocalized. It turns out that the vanishing of the phase rigidity indicates the delocalization transition.

Fig.2 (a) shows the phase diagram depending on the presence and absence of the $\Delta V$ effect on the state at $\theta = \pi/2$. Here the order parameter is given by $|\langle \sigma_z \rangle|$, where $\sigma_z$ is the Pauli matrix acting on the wave functions of the $A$ and $B$ sublattices in Eq.(6). Note that the order parameter $|\langle \sigma_z \rangle|$ measures the difference between the probabilities on the $A$ and $B$ sublattices. The yellow region indicates that the probability distributions for $A$ and $B$ atomic species are different due to the potential difference, $\Delta V$. In contrast, the blue region indicates that for each $k$ the probability distribution of $v_{\pm}(k)$ is completely uniform and independent of the atomic species. The phase boundary is given by the zero phase rigidity for each $k$ values. Note that when $\theta = 0$, the hermitian system, the blue region is impossible for any $k$ and finite $T$. This is because the non-reciprocal hopping phase could induce the delocalization of the states by interference effect instead of just increasing the mobility of the particle. Fig.2 (b) illustrates the order parameter, $|\langle \sigma_z \rangle|$ on the complex energy plane at $T = \Delta V$ and $\theta = \pi/2$. Remarkably, the zero values of $|\langle \sigma_z \rangle|$ appear only for the complex-valued energies. This is because the energy eigenvalues of the Hamiltonian, Eq.(5) are given by,

$$E_{k,\pm} = \frac{1}{2}\left(V_A + V_B \pm \sqrt{\Delta V^2 + 16t^2 \cos^2(ka/2)}\right). \tag{7}$$

Hence, at $\theta = \pi/2$, the energy becomes complex if and only if $T$ is larger than $\frac{\Delta V}{4\cos(ka/2)}$, which is the blue region of Fig.2 (a).

In terms of the hopping phase, one can explore different localization characteristics, not only exponentially localized or uniformly extended, but also power-law decaying critical states depending on $V_i$. To show the high controllability of the localization characteristics of the states, we consider the quasi-periodic systems in one-dimensional space. We study the Aubry-

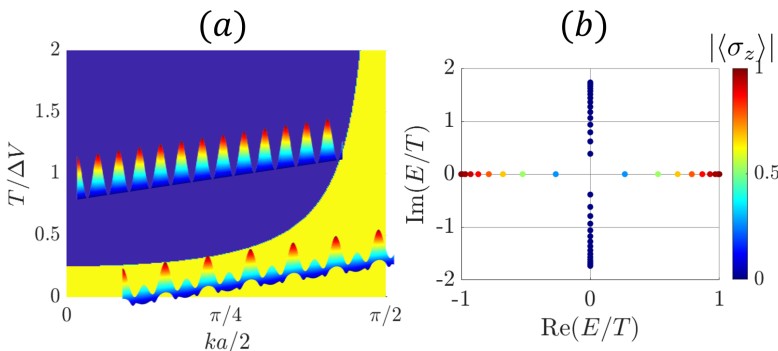

Figure 2: (a) State-dependent localization transition at $\theta = \pi/2$ for the alternating periodic chain. For each value of $k$, the blue region indicates that the wave function is uniformly distributed regardless of the atomic species, the same as for the uniformly periodic chain. The yellow region indicates that the wave function has skewed probability amplitude between $A$ and $B$ atoms. The probability distributions for each case are drawn schematically. For the Hermitian case, the blue region does not appear for finite $T$. (b) $|\langle \sigma_z \rangle|$ of the eigenstates on the complex energy plane at $\theta = \pi/2$. Here, we set $T = \Delta V$ and $V_A = -V_B$. The zero $|\langle \sigma_z \rangle|$, which indicates the uniformly distributed wave function, appears with the purely imaginary energies.

André Fibonacci model and discuss the effect of the non-Hermiticity induced by the non-reciprocal hopping phase. This model includes the Fibonacci quasicrystal limit, where both exponentially localized and critical wave functions exist. We emphasize that our discussion can be generalized to other systems or to the randomly disordered systems [73] (see Appendix A for detailed information).

## 3.1 Aubry-André-Fibonacci model

The Aubry-André-Fibonacci (AAF) model [67] is a 1D chain with quasi-periodically modulated $V_i$ given by,

$$V_i(\beta) = -\lambda \frac{\tanh[\beta \cos(2\pi \alpha i + \varphi) - \beta \cos(\pi \alpha)]}{\tanh \beta}. \tag{8}$$

Here $\varphi$ is the phase shift, indicating the global spatial translation of the potential, which we set to $\varphi = 0$. $\alpha$ is the golden section, $(1 + \sqrt{5})/2$. With respect to $\beta$, the model deforms continuously from the Aubry-André model [20] ($\beta \to 0$) to the Fibonacci limit [67] ($\beta \to \infty$), which we discuss in Sec.3.2. The Aubry-André model has been actively studied for the metal-insulator transition with respect to $T/\lambda$ in the incommensurately ordered systems [61, 63, 67]. Given $\beta$, we investigate the localization strength of the states as a function of the non-reciprocal phase, $\theta$ and $T/\lambda$.

Before illustrating the results, we briefly review the localization properties of the Hermitian AAF model under the open boundary condition (OBC). Under OBC, the Hermitian AAF model possesses at least one exponentially localized state regardless of finite $T$ and $\lambda \neq 0$. Thus, the maximally localized state in AAF models is exponentially localized regardless of $T/\lambda$ and $\beta$ [74]. For example, when $\beta = 0$, the fraction of localized states among the eigenstates changes drastically at the boundary of $T/\lambda = 1/2$ from 1 to $\sim 1/N$, where $N$ is the finite system size. This change is known as the metal-insulator phase transition, where the insulating and metallic phases refer to the $T/\lambda < 1/2$ and $T/\lambda > 1/2$ regimes, respectively [20]. For non-zero $\beta$, there are only a few delocalized states for $T/\lambda < 1/2$. For example, for $\beta = 2.5$, the delocalized state exists even for $T = 0.05\lambda$.

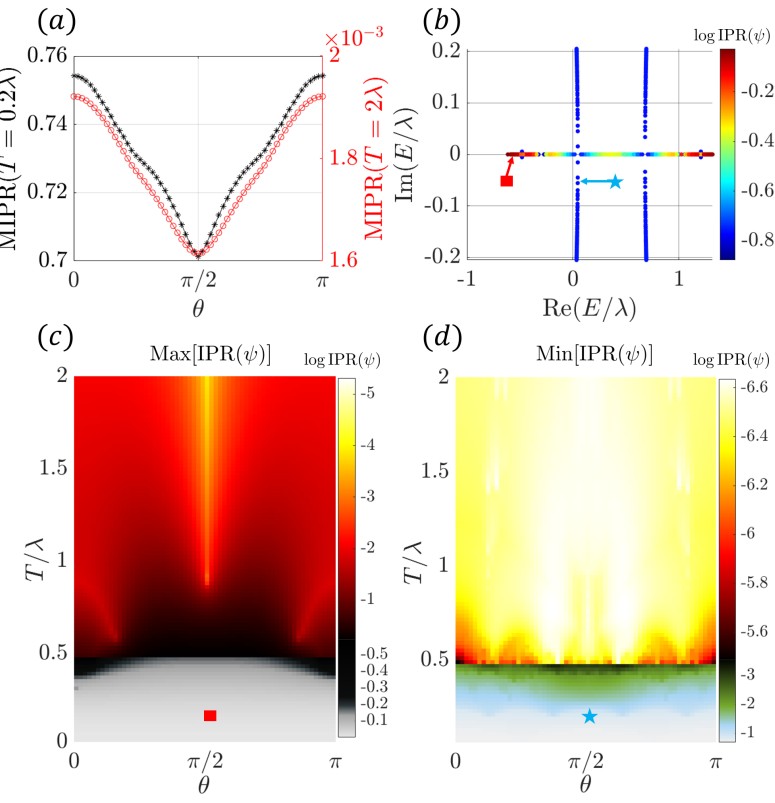

Figure 3: Changes of localization strength in the non-Hermitian AAF model for $\beta = 0$ as a function of $\theta$. $N = 987$. (a) Suppression of MIPR as $\theta$ approaches $\pi/2$, for both $T = 0.2\lambda$ (black curve) and $T = 2\lambda$ (red curve). (b) log(IPR) of the states on the complex energy plane at $T = 0.2\lambda$ and $\theta = \pi/2$. Smaller log(IPR) values appear mostly at the complex-valued energies. The red square and blue star indicate the maximally localized and maximally extended states in (c) and (d), respectively. Landscape of log(IPR) for (c) maximally localized state and (d) maximally extended state. For $T < 0.5\lambda$, (c) the localization strength of the maximally localized state (exponentially localized) increases in terms of IPR enhancement as $\theta$ gets closer to $\pi/2$. (d) The maximally extended state (exponentially localized) shows the decrease of the IPR as a function of $\theta$. In contrast, for $T \geq 0.5\lambda$, both the maximally localized (exponentially localized) and maximally extended (extended) states are delocalized as $\theta$ approaches $\pi/2$, as shown in (c) and (d).

Now we explore the non-Hermitian AAF model, where the non-Hermiticity is induced by the non-reciprocal hopping phase $\theta$. For different $\beta$, Fig.3 and Fig.4 show the IPR change and illustrate the localization and delocalization as functions of $\theta$. We emphasize that although the value of the IPR of an extended state would decrease with increasing system size, the qualitative change of the localization strength as a function of $\theta$ is independent of the system size. Here, we set $N = 987$. First, for $\beta = 0$, Fig.3 (a) shows that the MIPR decreases as $\theta$ approaches $\pi/2$ independent of $T$, and thus the delocalization of the state is generally observed. Fig.3 (b) shows each state on the complex energy plane and its IPR at $T = 0.2\lambda$ and $\theta = \pi/2$. Note that when the IPR of the state is relatively large, the energy of the state remains real-valued, while the energies of the states with smaller IPR values are mostly complex-valued. This is because the non-Hermiticity is driven by the non-reciprocal hopping phase, which does not affect the energy of strongly localized states. The maximum and minimum values of the IPR are emphasized as the red squares and blue stars, respectively (see Figs.3 (b-d)). Figs.3

(c) and (d) illustrate that $T < 0.5\lambda$ and $T \geq 0.5\lambda$ show different behavior in the localization strength for each state. In particular, for $T < 0.5\lambda$, the IPR of the maximally localized state increases as $\theta$ approaches $\pi/2$ as shown in Fig.3 (c). Whereas, Fig.3 (d) shows that, for $T < 0.5\lambda$, the IPR of the maximally extended state decreases as $\theta$ approaches $\pi/2$. Thus, the change in the localization strength varies for different states. This observation supports that the change of the localization strength is different for each state when the strength of the non-Hermiticity, $T \sin\theta$ is small. On the other hand, for $T \geq 0.5\lambda$, the non-Hermiticity generally leads to the delocalization of each state. Figs.3 (c) and (d) show that both the maximum and minimum values of the IPR decrease as $\theta$ approaches $\pi/2$. This indicates that the states are delocalized due to the non-reciprocal hopping phase. In particular, although the maximally localized state is still exponentially localized for $T \geq 0.5\lambda$ under OBC, its localization strength given by the IPR decreases as a function of $\theta$, unlike the case of $T < 0.5\lambda$.

Next we consider the case of non-zero $\beta$, exemplifying $\beta = 2.5$, which shows significant changes in the localization strength in $T < 0.5\lambda$ regime, compared to the case of $\beta = 0$. We observe the state localization with respect to $\theta$ when $T < 0.5\lambda$. The black curve of Fig.4 (a) shows that the MIPR for $T = 0.2\lambda$ is enhanced with non-trivial $\theta$. This enhancement of the MIPR is quite general for the $T < 0.5\lambda$ regime. In detail, Figs.4 (c) and (d) show the enhancement of the IPR of the maximally localized and extended states as $\theta$ approaches $\pi/2$ for the small $T < 0.5\lambda$ regime. In particular, Fig.4 (d) shows that the exponential localization tendency of maximally extended states with varying $\theta$ due to the non-reciprocal hopping phase. Specifically, the sky blue region in Fig.4 (d) does not admit the extended states. Note that if the maximally extended state becomes localized, then there is no extended state (see Appendix B for detailed information). Thus, each state is exponentially localized, similar to the case of $\beta = 0$. This is because when $T$ is small, the modulation of the potential distribution due to the hopping phase is weak, and hence the effective potential distribution resulting from the interference effect is similar to the case of $\beta = 0$ rather than uniformly periodic. Fig.4 (b) illustrates the log(IPR) on the complex energy plane for $T = 0.2\lambda$ and $\theta = \pi/2$. Here, the red square and blue star indicate the maximally localized and extended states. The strongly localized states have the real-valued energies, while the less localized states have the complex-valued energies. This supports that the localization for the $T < 0.5\lambda$ regime is strongly influenced by the non-reciprocal hopping phase. For $T \geq 0.5\lambda$, on the other hand, the red curve in Fig.4 (a) shows the delocalization of the states in terms of MIPR suppression as $\theta$ approaches $\pi/2$. Note that for the $T \geq 0.5\lambda$ regime, the delocalization of the states is a common signature with the $\beta = 0$ case (see Figs.4 (c) and (d)).

Comparing Fig.3 and Fig.4, one can conclude that the non-reciprocal hopping phase can either increase or decrease the localization strength depending on the state and the potential distribution dependent on $\beta$. In particular, Fig.3 (a) and Fig.4 (a) show that when $T = 0.2\lambda$, the change of the MIPR is distinct depending on $\beta$. When $T/\lambda$ is small, for $\beta = 0$, the localization strength decreases, while for $\beta = 2.5$, the localization strength increases as $\theta$ approaches $\pi/2$. This implies that the non-reciprocal hopping phase allow different control of the localization strength. Nevertheless, we find that when the hopping magnitude is sufficiently strong, the delocalization of the states is generally induced by decreasing MIPR for both $\beta = 0$ and $\beta = 2.5$ (see the red curves in Figs.3 (a) and 4 (a)). This is because the non-Hermitian interference effect which leads the effective potential to be uniform, and hence for $T$ greater than some critical hopping magnitude, say $T_c$, the interference in terms of $\theta$ uniformly washes out the effects of the potential gradient in the probability distribution, as we have also shown in the example of an alternating periodic chain.

The general tendency of state delocalization in AAF models for large $T$ could be also understood in terms of state hybridization between delocalized states. When $T$ is large, the Hermitian Hamiltonian possesses delocalized states whose probability distribution is locally

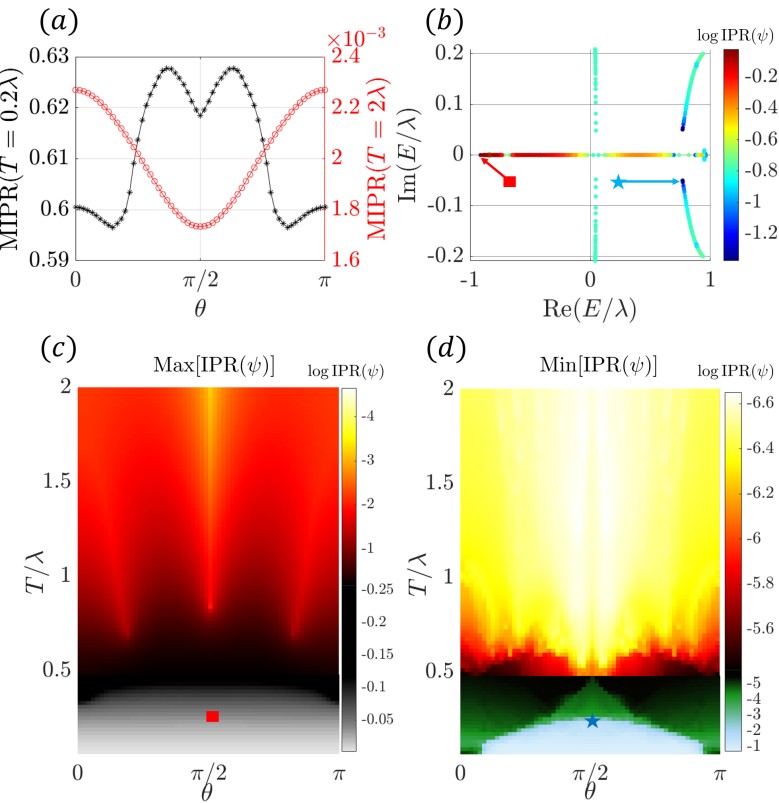

Figure 4: Changes in localization strength in the AAF model for $\beta = 2.5$ as a function of $\theta$. $N = 987$. (a) The MIPR for small hopping magnitudes increases for non-trivial $\theta$, as shown in black curve for $T = 0.2\lambda$. Whereas, it decreases monotonically for large hopping as $\theta$ approaches $\pi/2$, as shown in the red curve for $T = 2\lambda$. (b) log(IPR) of the states on the complex energy plane at $T = 0.2\lambda$ and $\theta = \pi/2$. The states with large log(IPR) values admit the real-valued energies, while the states with relatively small log(IPR) values have the complex-valued energies. The red square and blue star indicate the maximally localized and maximally extended states in (c) and (d), respectively. Landscape of log(IPR) for (c) maximally localized state and (d) maximally extended state. For $T < 0.5\lambda$, (c,d) the localization strengths of the maximally localized (exponentially localized) and maximally extended states increase in terms of IPR enhancement as $\theta$ approaches $\pi/2$. In particular, the sky region in (d) indicates the exponentially localized states, while the green and black regions in (d) indicate the extended states. In contrast, for $T \geq 0.5\lambda$, both maximally localized (exponentially localized) and maximally extended (extended) states become delocalized as $\theta$ approaches $\pi/2$, similar to the case of $\beta = 0$, as shown in (c) and (d).

non-uniform. Then, the non-reciprocal hopping phase induces the hybridization between these delocalized states, which compensates the local difference of probability amplitudes. For example, in the case of an alternating periodic chain, the difference of the probability amplitude at the $A$ and $B$ sublattices in $|v_+(k)\rangle$ and $|v_-(k)\rangle$ is compensated as $\theta$ approaches $\pi/2$. In particular, after the coalescence of $|v_+(k)\rangle$ and $|v_-(k)\rangle$ at $\theta = \pi/2$, the local probability difference between $A$ and $B$ sites is fully compensated. The coalescence of states is a unique feature of the non-Hermitian system [69, 75]. Thus, such anomalous state hybridization is another key mechanism leading to the delocalization of states. Moreover, this kind of state delocalization is also observed from the weak localization in randomly disordered systems. See Appendix A for detailed information.

To summarize, the non-reciprocal hopping phase can control the localization strength of the states in AAF models in different ways, either increasing or decreasing the localization strength. This opens up new experimental applications of AAF models. For example, based on the result of the case $T < 0.5\lambda$ for $\beta = 2.5$, one can control the localization characteristics from the mixture of extended and localized states to the perfectly localized spectrum. On the other hand, when $T > 0.5\lambda$, the non-Hermiticity induces the delocalization of the states. Thus, depending on the hopping magnitude, one can either promote or suppress the mobility of the particle as a function of the non-reciprocal hopping phase.

### 3.2 Fibonacci quasicrystal

Now, let us consider the limit $\beta \to \infty$ of the AAF model, the Fibonacci quasicrystal [31,76]. In detail, the Fibonacci quasicrystal consists of two different atoms, $A$ and $B$, which have onsite potentials $V_A$ and $V_B$, respectively. From Eq.(8), we set $V_A = \lambda$ and $V_B = -\lambda$. By using the successive substitution maps, $A \to AB$ and $B \to A$, one obtains the Fibonacci arrangement of the atoms, such as $ABAABABABAA\cdots$ [12, 25]. For the Hermitian system under OBC, it is known that there are both exponentially localized states and critical states in the Fibonacci quasicrystal, independent of the finite hopping magnitude, $T$ [62, 77, 78]. Thus, one could ask whether non-Hermiticity with a uniform complex-valued hopping parameter enhances the delocalization of the states in the Fibonacci quasicrystal and eventually gives rise to the extended states for a finite hopping magnitude under the OBC.

To quantify the localization characteristics of the wave function in the Fibonacci quasicrystal, we use both the IPR and the fractal dimension of the state. Although a larger IPR indicates stronger localization, the value of the IPR alone is not sufficient to determine the detailed localization characteristics and scaling behavior of the wave function, especially the critical state, since the IPR is an averaged quantity over space [16,71]. Note that for the case of finite $\beta$, the localization properties are already well explained by the IPR, since we do not have critical states and phase transitions exploring unconventional fractal dimensions (see Appendix B for detailed information). However, for the Fibonacci quasicrystal given by $\beta \to \infty$, the critical state would appear with intermediate localization characteristics and unique spatial distribution, such as power-law scaling behavior. Therefore, a more detailed quantification of the different localization characteristics is required. Thus, we study the system size dependence of the IPR, which gives the spatial distribution of the wave function. In particular, it is known that for sufficiently large system size $N$, the IPR exhibits scaling behavior as $N^{-D_2}$, where $D_2$ is called the fractal dimension [16,21,79]. An exponentially localized state has $D_2 = 0$ and a uniformly extended state has $D_2 = 1$. The critical states have the intermediate fractal dimensions $0 < D_2 < 1$ [16,79].

Figs.5 (a) and (f) show the landscapes of the fractal dimension of the maximally localized state and the maximally extended state, respectively, as a function of the magnitude and phase of the hopping parameter $T$ and $\theta$ (see Appendix D for detailed information on the fractal dimension). The potential difference between atoms $A$ and $B$ is given by $V_A - V_B = 2V$ and $V = \lambda$ from Eq.(8). As $\theta$ approaches $\pi/2$, the fractal dimensions of the maximally localized or maximally extended states increase, indicating the delocalization of the states. In detail, Fig.5 (b)-(d) show that for sufficiently large $T$ ($T > 10V$ in Fig.5 (a)), how the localization characteristics of the maximally localized state are controlled from the exponentially localized to the sinusoidally extended state with respect to $\theta$. In the Hermitian case, the fractal dimension of the maximally localized state remains zero, corresponding to the exponentially localized state, regardless of the value of $T$ (see Figs.5 (a) and (b)). However, in the non-Hermitian cases, the fractal dimension of the maximally localized state becomes non-zero. Fig.5 (c) shows the wave function of the maximally localized state for $\theta = 17\pi/36$ with $T = 13V$ (red square in Fig.5 (a)). In this case, the wave function shows the critical state with a power-law decay, and

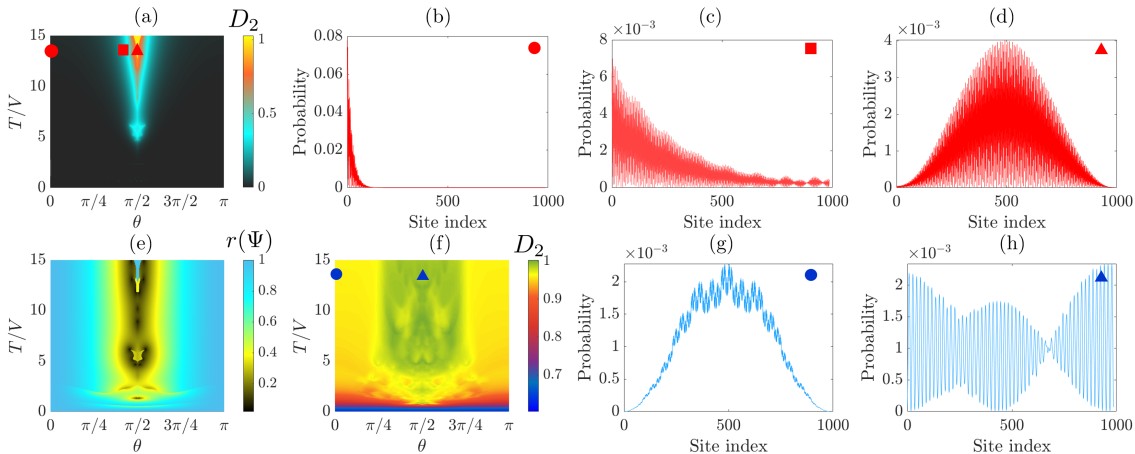

Figure 5: Change of the localization characteristics of the states in the Fibonacci quasicrystal. The landscape of fractal dimensions of the states for (a) maximum value of IPR and (f) minimum value of IPR, corresponding to the maximally localized and maximally extended state, respectively. (b-d) Evolution of the localization characteristics of the maximally localized state for (b) $\theta = 0$, (c) $\theta = 17\pi/36$ and (d) $\theta = \pi/2$, respectively, with $T = 13V$. $V = (V_A - V_B)/2$ is the difference between two kinds of the on-site energies, $V_A$ and $V_B$ in the Fibonacci quasicrystal model. Here, $V_A = 1$ and $V_B = -1$. The fractal dimensions are (b) $D_2 = 0$, (c) $D_2 = 0.411$ and (d) $D_2 = 1$. As the phase approaches the maximum value of non-Hermiticity at $\theta = \pi/2$, the localization characteristics of the states change from the exponentially localized state (b) to the extended state (d), where intermediate power-law decaying critical states (c) appear at intermediate $\theta$. (e) The phase rigidity of the maximally localized state. Along the boundary where the fractal dimension changes rapidly in panel (a), the phase rigidity also drops sharply in the panel (e). (g-h) Evolution of the localization characteristics of the maximally extended state for (g) $\theta = 0$ and (h) $\theta = \pi/2$, respectively, with $T = 13V$. The fractal dimensions are (g) $D_2 = 0.915$ and (h) $D_2 = 1$. The localization characteristics change from the self-similar critical state (g) for the Hermitian case to the uniformly oscillating extended state (h) for the maximally non-Hermitian case. The wave functions are plotted for the system size, $N = 987$. The fractal dimension is calculated using the system sizes from $N = 300$ to $N = 987$. See the main text for further details.

the fractal dimension $D_2 = 0.411$. Moreover, Fig.5 (d) shows the perfectly delocalized wave function with $D_2 = 1$ for $\theta = \pi/2$ with $T = 13V$ (red triangle in Fig.5 (a)). It is surprising that for sufficiently large but finite hopping magnitude, the maximally localized state becomes an extended state with $D_2 = 1$. In this case, every eigenstate has $D_2 = 1$, which is never allowed in the Hermitian system for any finite hopping magnitude $T$ and non-zero $V$.

The reason why we can universally achieve $D_2 = 1$ in the Fibonacci quasicrystal even with finite hopping magnitude is as follows. The non-reciprocal hopping phase controls the localization of the states by forming a non-trivial interference between the hopping and the potential contributions of the state, which arise from the hopping parameters and the spatial potential gradient, respectively. If $T$ is sufficiently large, the interference becomes destructive as $\theta$ approaches $\pi/2$. Consequently, the potential contribution can be canceled out by the hopping contribution. As a result, the non-Hermiticity leads to the delocalization of the states by effectively blinding the spatial potential gradient such as the quasiperiodic structures with the destructive interference. This allows us to achieve the uniformly extended state even in the presence of the spatial potential gradient with finite hopping magnitudes.

Although the maximally localized state remains exponentially localized as $D_2 = 0$ regardless of $\theta$ when $T$ is small ($T \lesssim 4V$ in Fig.5 (a)), the non-Hermiticity increases the localization length of the state. It turns out that the interference arising from the non-reciprocal hopping phase induces the penetration of the wave function into the bulk of the system. Thus, even with a small $T$, we can manipulate the localization strength of the state with respect to the non-reciprocal hopping phase. See Appendix C for detailed information on controlling the localization length.

The strong state hybridization and coalescence are also important for understanding the change in the scaling properties of the state. To capture this, we compute the phase rigidity, $r(\Psi)$ of the maximally localized state. Recall that $r(\Psi) = 1$ for the Hermitian case, while $r(\Psi) \leq 1$ for the non-Hermitian case, because the right eigenstates may not be orthonormal to each other. Fig.5 (e) shows the landscape of phase rigidity. Comparing Figs.5 (a) and (e), the phase rigidity suddenly drops at the boundaries where the change in localization characteristics occurs. Thus, when the localization characteristics change, the maximally localized state strongly hybridizes with other critical or extended states by passing through the exceptional point [69]. This leads to delocalization of the scaling behavior of the maximally localized state.

Now let us focus on the maximally extended state. In the Hermitian system, the maximally extended state has the fractal dimension $D_2 < 1$ for any finite $T$ due to the fractal structure of the Fibonacci quasicrystal [77]. However, Fig.5 (f) shows that the non-Hermiticity can increase the fractal dimension of the maximally extended state as $D_2 = 1$. Comparing the wave functions for the maximally extended states at $\theta = 0$ (Fig.5 (g)) and $\pi/2$ (Fig.5 (h)), one can see *the disappearance of the fractality* in the wave function due to the non-Hermiticity. It turns out that the non-reciprocal phase factor of the hopping parameters induces delocalization by shielding the detailed structure of the lattice, such as the Fibonacci pattern, through the interference effect, rather than simply increasing the mobility of the particle.

One can ask how the localization strength of the other states changes as the strength of the non-Hermiticity increases. For a given finite $T \geq 0.2V$, we generally see that the MIPR decreases as $\theta$ approaches $\pi/2$ in the Fibonacci quasicrystal (see Fig.6). Thus, non-Hermiticity leads to the delocalization of states. In detail, Fig.6 shows the MIPR as a function of the hopping phase, $\theta$, for different hopping magnitudes $T$ in the Fibonacci quasicrystal. For any $T$, the MIPR decreases as $\theta$ gets closer to $\pi/2$. Thus, the localization strength in the spectrum is suppressed in the Fibonacci quasicrystal due to the non-Hermiticity. Note that in the Fibonacci quasicrystal, even for the small $T$ regime, most of the states are critical states, which are delocalized as a power-law scaling. Thus, the hybridization of the eigenstates of the Hermitian Hamiltonian due to the non-reciprocal hopping phase occurs mainly between pairs of critical states in a way that compensates for the power-law decaying probability amplitudes. This leads to the general delocalization tendency in the presence of the non-reciprocal hopping phase.

## 4  Discussion and Conclusion

Over the past decade, advances in photonics and electronics have made it possible to experimentally realize open systems, described as various effective non-Hermitian Hamiltonians. However, how non-Hermiticity controls localization properties of the wavefunctions, is hardly understood. In this work, we show that the localization of wavefunctions, including quasi-periodic structures and disordered chains, is tuned in non-Hermitian system with non-reciprocal hopping phase. We show that the non-Hermiticity imposed by the non-reciprocal hopping phase does not cause a skin effect, but can dramatically change the localization of the wave function, even inducing general delocalization with sufficiently strong non-Hermiticity. While the non-Hermitian skin effect strongly depends on the boundary conditions, the delocal-

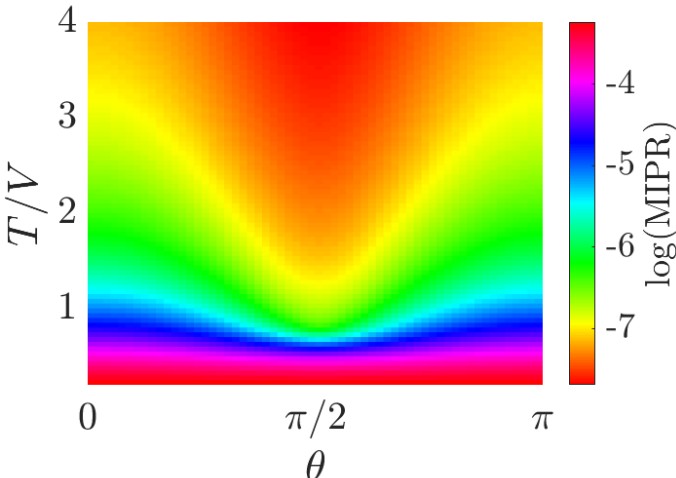

Figure 6: The mean value of the IPR (MIPR) of the energy spectrum as a function of the strength of the non-Hermiticity given by the hopping phase, $\theta$ in the Fibonacci quasicrystal model. At $\theta = \pi/2$, the non-Hermiticity becomes maximum for given $T/V$, where $T$ is the hopping magnitude and $V = (V_A - V_B)/2$ is the difference between two kinds of the on-site energies, $V_A$ and $V_B$ in the Fibonacci quasicrystal model. Here, $V_A = 1$ and $V_B = -1$. The MIPR (Eq.(4)), which is the amount of the localization in the spectrum decreases as the non-Hermiticity becomes stronger. $N = 987$.

ization induced by the non-reciprocal hopping phase does not depend on any specific boundary conditions. We show that there are two main mechanisms by which the non-reciprocal hopping phase changes the localization properties of the wave function.

The first mechanism is interference between transition probability amplitudes. The non-reciprocal hopping phase imposes a nontrivial phase difference between the transition probability amplitudes given by different paths. This nontrivial phase causes destructive interference between the transition probability amplitudes, which is impossible in the Hermitian system, and causes the eigenstates to spread further in space. The effect of this interference grows with increasing magnitude of the hopping parameter, as well as with non-reciprocal hopping phases, and leads to the delocalization of the wave function.

Another mechanism is the coalescence of the eigenstates through the exceptional points. In terms of phase rigidity, we clarify that the localized (either exponentially or critically) state merges into the delocalized state at the boundary points where the localization characteristics given by the fractal dimension are drastically changed [cf. comparison between Figs.5 (a) and (e)]. Thus, the states are delocalized when passing through the exceptional point. Remarkably, this is different from the case of the skin effect, where the presence of the exceptional point leads to macroscopic localization. Thus, our study extends the role of non-Hermitian exceptional points from the traditional skin effect to drastic changes of the fractal dimension of the states.

We provide a novel way to control the localization of quantum states by exploring non-Hermiticity. Our non-Hermitian model, whose non-Hermiticity is originated from the non-reciprocal phase of the hopping parameter, has no specific directional preference. It is important to note that, depending on the state, the localization can be enhanced or reduced as a function of the non-reciprocal hopping phase and the magnitude of the hopping parameter. In this way, one can finely control the localization of the states. In particular, for exponentially localized states or critical states present in quasicrystalline systems, the non-Hermiticity

induces a perfect delocalization of the states, resulting in the disappearance of the fractality. Using the Fibonacci quasicrystal as an example, we have shown that non-Hermiticity can indeed change the localization properties between localized, critical, and uniformly extended states. Again, this is due to the interference between the transition amplitudes with respect to the non-reciprocal hopping phase and the hybridization of the states by the exceptional points, which compensate for the non-uniform amplitudes of the probabilities and lead to the delocalization of the states in the regime of strong non-Hermiticity. Our work opens the utility of non-Hermiticity for high controllability of the localization properties.

Our theoretical work could be experimentally studied in the photonic crystal [43] or electrical circuits similar to other open system models governed by the Lindblad master equation or the effective non-Hermitian Hamiltonian. In particular, we propose an experimental setup to demonstrate the control of the localization of the wave function in the quasiperiodic system as shown in Fig.1. By changing the phase accumulated by the phase shifter, one can explore the different localization characteristics of the wave functions from the exponentially localized to the critical or extended states. The electric circuit or the acoustic lattice are also possible platforms to demonstrate the localization control in terms of the non-reciprocal phase of the transporting waves [60].

Beyond the one-dimensional systems we considered here, one can generalize our model to higher-dimensional systems such as two- or three-dimensional lattices [79–82], or even to the multi-frequency driven Floquet systems with synthesized dimensions [83,84]. We suggest that similar delocalization phenomena would occur in the higher-dimensional lattices due to the non-reciprocal phases of the hopping parameter. In such a case, the state-dependent control of the localization strength and the emergence of subdimensional fractality can also be discussed, which we leave as an interesting future work.

## Acknowledgsments

We thank Yidong Chong and Dung Nguyen Xuan for useful discussions.

**Funding information**   J.M.J. and S.B.L. are supported by National Research Foundation Grant (No. 2021R1A2C1093060), National Research Foundation Grant (RS-2023-00281839).

## A   Control of localization in the uniformly random disordered chain

Here, we consider the randomly disordered chain, where the on-site energies have a level of disorder of 50%. Specifically, the random on-site energies are between $-1.5V$ and $-0.5V$. Fig.7 shows that the localization is suppressed as the non-Hermicity increases, even for the randomly disordered chain. The MIPR decreases as $\theta$ becomes $\pi/2$, which corresponds to the maximum strength of the non-Hermiticity for a given hopping magnitude $T$. Thus, the delocalization is induced by the non-reciprocal hopping phases in the randomly disordered chain.

## B   Localization of the extended states for the AAF model with $\beta = 2.5$ and small $T$

Here, we discuss the localization of the extended states for $\beta = 2.5$ and small $T$. In particular, we focus on the maximally extended state, which has the minimum value of the IPR in the

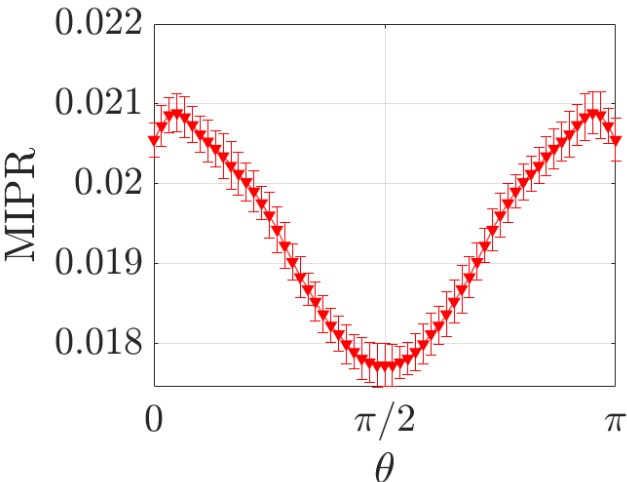

Figure 7: MIPR as a function of the phase of the non-Hermitian hopping parameter, $\theta$, in the randomly disordered system. The degree of disorder of the on-site potential energy is 50%. The system size, $N = 233$, and the hopping parameter value, $T = 4V = 4$. The number of samples is 20.

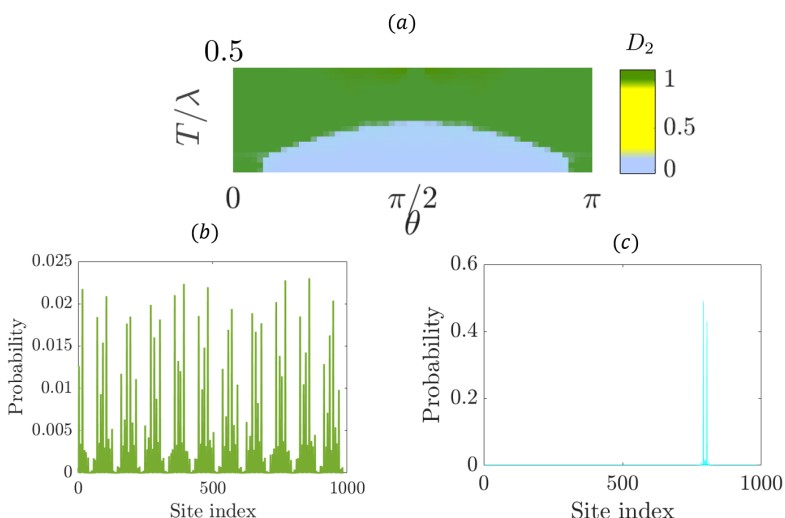

Figure 8: Localization of the maximally extended state for $\beta = 2.5$ and $T < 0.5\lambda$. (a) Landscape of the fractal dimension of the maximally extended states. The fractal dimensions are either 0 or 1. The intermediate fractal dimension, colored yellow, does not appear. (b) The extended state for $\theta = 0$ and $T = 0.2\lambda$. (c) The localized state for $\theta = \pi/3$ and $T = 0.2\lambda$. The wave functions are plotted for the system size, $N = 987$. The fractal dimension is calculated using the system sizes from $N = 300$ to $N = 987$.

spectrum. Note that if the maximally extended state becomes a localized state, then every state in the spectrum is localized, i.e. the extended states disappear in the spectrum.

Fig.8 shows the exponential localization of the extended states for the AAF model with $\beta = 2.5$ and $T < 0.3\lambda$. Fig.8 (a), which shows the landscape of the fractal dimension $D_2$ of the maximally extended state, emphasizes the absence of the extended state in the spectrum as the sky blue region ($D_2 = 0$, exponentially localized). The green region in Fig.8(a) indi-

cates the extended states ($D_2 = 1$). Note that the critical states characterized by $0 < D_2 < 1$ (yellow in the color bar of Fig.8 (a)) do not appear. Figs.8 (b) and (c) show the typical wave functions corresponding to the green and sky blue regions of Fig.8 (a), respectively. Thus, as $\theta$ approaches $\pi/2$, each eigenstate becomes exponentially localized, and the extended states disappear from the spectrum. Again, we emphasize that there is no critical state during the localization transition due to the non-reciprocal hopping phase.

## C  Localization length change in the Fibonacci quasicrystal with small hopping magnitude regime

Here, we consider the Fibonacci quasicrystal with the small hopping magnitude, $T$ regime. The localization length is controlled by the strength of the non-Hermiticity, although the fractal dimension is always zero for the maximally localized state with the maximum value of the IPR in the small $T$ regime. Specifically, we define the localization length, $\xi$, of the exponentially localized state as follows

$$\xi = \sqrt{\langle \hat{x}^2 \rangle - \langle \hat{x} \rangle^2}, \tag{C.1}$$

where $\hat{x}$ is the position operator and $\langle \hat{O} \rangle$ is the expectation value of the operator, $\hat{O}$. For $T = 3V$ we show the variance of the localization length as a function of the non-reciprocal phase of the hopping parameter, $\theta$. Recall that the uniform non-Hermitian hopping parameter is given by $t = Te^{i\theta}$.

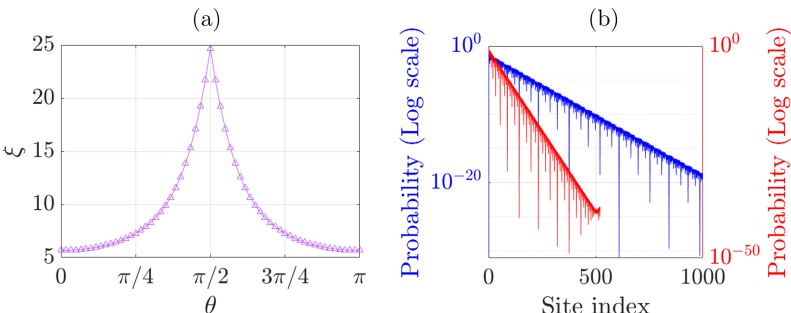

Figure 9: (a) The localization length ($\xi$ defined in Eq.(C.1)) changes as a function of the non-reciprocal hopping phase ($\theta$) in the Fibonacci quasicrystal model. The unit of the localization length is the atomic distance between neighboring atoms, which is set to 1. The non-Hermiticity induces the delocalization, so the localization length increases as the non-Hermiticity becomes stronger. (b) Comparison of the probability distribution of the maximally localized states for (blue) $\theta = \pi/2$ and (red) $\theta = 0$ in the logarithmic scale. The linear scaling in the figure shows the exponential decay. The smaller slope indicates the larger localization length for the non-Hermitian case. The hopping magnitude is $T = 3V = 3$. The system size is $N = 987$.

Fig.9 (a) shows that the localization length increases drastically as $\theta$ approaches $\pi/2$. Thus, although the localization properties of the maximally localized state are exponentially decaying for any $\theta$ with small $T$, the localization length could be manipulated in terms of the non-reciprocal hopping phase. Fig.9 (b) compares the probability distribution of the maximally localized states for $\theta = 0$ (red) and $\theta = \pi/2$ (blue) on the logarithmic scale. The different slopes show that in the non-Hermitian case the wave is able to penetrate further into the bulk.

# D   Detailed calculation of the fractal dimensions

Here, we present the detailed numerical calculations of the fractal dimensions. We consider finite system sizes from $N = 300$ to $N = 987$ to extract the power of the scaling behavior of the IPR. For each $N$ value, we calculate the IPR of the maximally localized and extended states, respectively. In Fig.10 we show the numerical results of the fractal dimensions of the maximally localized and extended states shown in Figs.5 (b,c,d,g,h). In detail, we compute the fractal dimension of the wave function by applying the linear regression method to $-\log(\text{IPR})$, which is the function of $\log(N)$. The IPR of the localized state would not depend on the system size. On the other hand, the IPR of the critical or extended states would decrease with increasing system size. The fractal dimension can be extracted from the slope of the regression line. The standard error of the fractal dimension is given by the standard error of regression slope. Note that the numerical errors are small enough to distinguish the localized, critical and extended states.

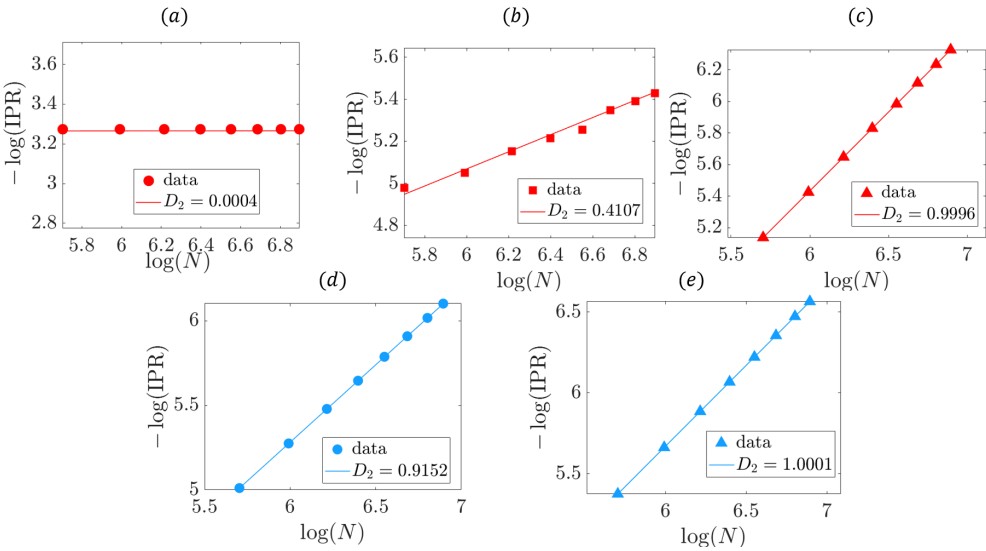

Figure 10:    Numerical calculation of the fractal dimension with standard error range.    The markers correspond to the states illustrated in Figs.5 (b,c,d,g,h).    (a) Localized state.    $D_2 = 0.0004 \in (-0.0001, 0.0009)$.    (b) Critical state.    $D_2 = 0.4107 \in (0.4009, 0.4205)$.    (c) Extended state. $D_2 = 0.9996 \in (0.9990, 1.0002)$. (d) Critical state. $D_2 = 0.9152 \in (0.9148, 0.9156)$. (e) Extended state.    $D_2 = 1.0001 \in (0.9998, 1.0004)$.    Here, $N = 300, 400, 500, 600, 700, 800, 900$, and $987$.  For information on the other parameters, see the main text.

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
