# Peer review of "Localization control born of intertwined quasiperiodicity and non-Hermiticity"

_SciPost Physics, doi:SciPost Phys. Core 6, 077 (2023)_

## Round 1 · Referee Report · Anonymous (Referee 1) · 2023-7-23

Strengths

  1. The manuscript studies a new model of non-Hermitian quasicrystal (non-Hermitian extension of the Aubry-André-Fibonacci model).

  2. The manuscript is written in an accessible manner.

Weaknesses

  1. More thorough analytical and numerical investigations should be needed to understand the phase transitions and critical behavior in the authors’ model, although they may go beyond the aim and scope of the present manuscript.

Report

The authors introduce and study a new model of non-Hermitian quasicrystal. Specifically, they introduce a non-Hermitian extension of the Aubry-André-Fibonacci model and study its inverse participation ratio and fractal dimension. The authors find unique behavior that has no counterparts in the corresponding Hermitian quasicrystal, such as the unconventional delocalization of the originally critical and localized states. Through these analyses, the authors claim that the localization properties of quasiperiodic systems can be nontrivially controlled by engineering non-Hermiticity, which may be of experimental relevance.

Recently, the physics of non-Hermitian Hamiltonians has attracted growing interest. One of the major current focuses is the interplay of disorder and non-Hermiticity. In this context, I believe that this manuscript, which faithfully studies a new model of non-Hermitian quasicrystal, can make a significant contribution in the research fields of non-Hermitian physics, and I would like to recommend publication of this manuscript in SciPost Physics.

Requested changes

I only have a couple of relatively minor comments, as explained below.

  1. In Sec. 3, the authors call “$r \left( k \right) = \left| \langle v_{+} \left( k \right)L | v \left( k \right) _R \rangle \right|$” the phase rigidity. While I understand that this quantity can detect the chaotic or localized behavior of eigenstates [e.g., N. Hatano and D. R. Nelson, Phys. Rev. B 58, 8384 (1998); J. T. Chalker and B. Mehlig, Phys. Rev. Lett. 81, 3367 (1998)], I fail to clearly understand why the authors call it the phase rigidity. I would appreciate clarification and justification.

  2. The manuscript contains some typographic or grammatical errors, which the authors should correct carefully. For example, above Eq. (5), “Remarkably, the zero values of $\left| \langle \sigma_z \rangle \right|$ appears only for the complex valued energies” should be “Remarkably, the zero values of $\left| \langle \sigma_z \rangle \right|$ appear only for the complex valued energies”. In the second last paragraph of Sec. 3.2, “… due to the non-Hermitianity” should be “… due to the non-Hermiticity”.

  • validity: good
  • significance: ok
  • originality: ok
  • clarity: ok
  • formatting: reasonable
  • grammar: reasonable

Author:  SungBin Lee  on 2023-08-10  [id 3891]

(in reply to Report 1 on 2023-07-23)
Category:
answer to question
correction

We appreciate the referee for giving a positive comment on our present manuscript.

Author response 1.
The terminology of the “phase rigidity” is widely used in the non-Hermitian physics. This measures the rigidity of the phase of eigenfunction. In the non-Hermitian system, the eigenfunctions are biorthonormal i.e.
$$\langle\psi_n^L|\psi_m^R\rangle=\delta_{nm}, $$ where $$\vert\psi^R\rangle=\frac{\vert\phi^R\rangle}{\sqrt{\langle\phi^L|\phi^R\rangle}} \ \mbox{and} \ \vert\psi^L\rangle=\frac{\vert\phi^L\rangle}{\sqrt{\langle\phi^L|\phi^R\rangle}} $$
Here, $R$ and $L$ stand for the right and left eigenstates, $H\vert\phi^R\rangle=E\vert\phi^R\rangle$ and $\langle\phi^L\vert H=E\langle\phi^L\vert$, where $H$ is the non-Hermitian Hamiltonian. $n,m$ are indices of the eigenstates.
However, at the exceptional point (zero phase rigidity), where two states coalescence, the states become linearly dependent, and the biorthonormal condition is failed. At this point, their phase (in a sense of the complex number) jumps---which is known as the characteristics of the branch point. Due to this jump of the phase, the phase of the eigenfunction is not rigid at vicinity of the exceptional points. Thus, we call it as the phase rigidity. Further details are explained in the review paper, [Ingrid Rotter 2009 J. Phys. A: Math. Theor. 42 153001].

Author response 2
We appreciate for pointing out the typos and grammar mistakes. We check our manuscript carefully and correct such typos and grammatical errors.

---

## Round 1 · Referee Report · Anonymous (Referee 2) · 2023-8-6

Report

The authors explore the wave-function localization in the Aubry-André-Fibonacci model where non-reciprocal hopping phases are present, i.e., the non-Hermiticity is encoded in the phases of the hopping terms. The authors find that the interplay between quasi-periodicity and the complex hopping amplitudes can lead to the perfect delocalization of the states.

I have to confess that I am not an expert in the field and thus have some difficulty assessing the relevance of the manuscript, i.e., I am not sure if it is really suitable for SciPost Physics proper or should rather go into SciPost Physics Core.

A possible experimental realization is emphasized both in the abstract and in Fig. 1. However, I could not help the impression that this is only an afterthought that the authors use to sell their work. It is not clear to me if this is experimentally feasible (are experimental colleagues really able to propagate light only in one direction?) and in any case, no experiments are presented in the present manuscript. I thus think that emphasis should be removed from possible experimental realizations.

The work itself appears to be solid. The presentation is sometimes nice, but at other places also difficult to follow. One issue is that the structure of the manuscript is not always optimal. For example, section 2 and the appendices A-C are very short. I recommend specifically:
a) Expand section 2 by the basic definitions. For example, the rigidity $r$ is introduced twice, once on page 6 and once on page 13, but both times only in the text. Collecting such definitions once at the beginning would remove redundancy and could improve clarity.
b) Move the appendices into the main text. For example, appendices A and C have only 7 lines of text such that moving them into the main text might improve readability. Of course, in particular in the case of the material presented in appendix A, relevant quantities would have to be defined first, but see suggestion a).

The "return probability" is introduced at the beginning of section 3, but not used in later analysis. I thus recommend to remove this definition and the related discussion, and rather use the notions that appear later for the main results also at the beginning of section 3.

The model Eq. (3) with $\theta=0$ would be the textbook example of a dimerized chain. It might helpful to add this term to the discussion.

I have a few further more specific comments that I list under "Requested changes".

Requested changes

1- Remove emphasis of the proposed experiments. 2- Collect all relevant definitions (rigidity, inverse participation ratio, ...) in section 2. 3- Move appendices into the main text. 4- Remove the "return probability" Eq. (2) and related text. 5- Mention that Eq. (3) is also known as a "dimerized chain" (?). 6- The abbreviation "NHSE" is only used in the second paragraph of the Introduction. I recommend avoiding the use of abbreviations that are only needed once. 7- Fig. 5: The meaning of the parameter $V$ is not clear; in particular no $V$ appears in the model Eq. (6). I admit that a $V$ is introduced in the text of the third paragraph of section 3.2, but the relation to Eq. (6) is still unclear. Please clarify. 8- Again Fig. 5: Better insert later such that it does not appear before it is discussed. 9- Caption of Fig. 5: The authors probably mean "panel (a)" and "panel (e)" rather than "Fig. (a)" and "Fig. (e)", respectively. 10- Second paragraph of page 13: I believe that the authors mean "non-orthonormal" rather than "non-orthogonal". 11- The abbreviation "EP" appears on page 13, but might actually not have been introduced. 12- Fig. 7: It is not clear from the caption what this figure shows. Please specify the model and relevant parameters in the caption. 13- Fig. 8: It is again not clear from the caption what this figure shows. Please specify the model and relevant parameters in the caption. 14- References: a) Always provide a DOI when possible, in particular for Refs. [6-11,13,35,38,41,42,53,57,60] b) Avoid spurious lowercasing of names in titles. In particular, "Aubry", "André", "Fibonacci", and "Anderson" should not be lowercased. c) It seems that Ref. [20] is published in Ann. Israel Phys. Soc. 3, 18 (1980). Given that this reference is central to the present work, the authors should make an effort to provide full details. d) It is not necessary to provide a link to the PDF if a DOI is specified (Refs. [33,55,56,76]). e) It seems that Ref. [79] is vol. 758 of the "Memoirs" of the American Mathematical Society. Actually, at first sight, it was not clear to me if this is a journal article with incomplete data or a book. I think that it would really help if the authors could present this more clearly. 15- The English is generally good, but sometimes there are minor grammatical errors or strange constructions. Let me just mention the "study" that could be "studied" in the last paragraph of the Discussion and Conclusion and the "drastically controlled" on line 2 of appendix B (I do not think that one can say this in English).

  • validity: high
  • significance: good
  • originality: good
  • clarity: good
  • formatting: good
  • grammar: good

Author:  SungBin Lee  on 2023-08-10  [id 3892]

(in reply to Report 2 on 2023-08-06)
Category:
answer to question
correction

Author response 1.

We thanks for the detailed review of our manuscript. Let us briefly emphasize the novelty and importance of our work. Recently, the non-Hermitian physics is widely studied in physics of the open systems such as optics and condensed matter physics. One of the major stream of the research in the non-Hermitian physics is the interplay of disorder and non-Hermiticity. However, the non-Hermitian system with non-reciprocal hopping phases has never been explored, and its interplay with the disorder or quasiperiodic order has been elusive. In this context, our findings of the phase transition from localized phase to delocalized phase due to the non-reciprocal hopping phases on the quasiperiodic system can give an important contribution in the general research areas related to the non-Hermitian physics and open systems. Hence, we believe that our work is suitable for SciPost Physics.

Author response 2. We agree with the comment on our Fig. 1 and abstract raised by the referee. Thus, we edited our manuscript in the section where we discuss the possible experimental applications and noted that Fig.1 is the potential experimental application.

The referee asked if it is really possible to propagate light only in one direction. This is possible by using the optical isolator (or sometimes called optical diode), which is generally used to study the non-reciprocal optics. An optical isolator is a device that allows light to propagate through it only in one direction, but not in the opposite direction. Isolators are useful as valves that allow propagation in only one direction. They are used in high-power applications, for which one desires one-way transmission of light. [Refer to the book, ``Encyclopedia of Physical Science and Technology'', ISBN: 978-0-12-227410-7]. We have added this reference, in the section we introduce the optical isolator. Hence, we believe that our suggestion of the possible experiment realization shown in Fig. 1 would be reasonable, and we are currently in communication with experimentalists in this field.

Author response 3 Based on the referee's comments, we have edited our manuscript for better understanding. Specifically, we expand the section 2 with including the basic definitions of the phase rigidity and inverse participation ratio (IPR). We also removed the redundant expressions in our manuscript. We also move the appendix C to the main text for better understanding. The relevant quantities such as mean inverse participation ratio (MIPR) are also introduced in section 2. We leave the appendix A as the appendix because it would interrupt the flow of the main text. We specify the model (random disordered chain or the Fibonacci chain), and added detailed numerical values in the captions of the figures.

Author response 4 As the referee has pointed out, the definition of the ``return probability'' in Eq.(2) has not been directly used except specifying the meaning of the interference due to the non-reciprocal hopping phases. Thus, based on the referee's comment, we remove the definition (Eq. (2)) and the related detailed discussion. Instead, we briefly explain how the coalesence of the states would change the localization characteristics of the states, which is one of the important notions that appear to explain our results.

Author response 5 The model Eq. (3) with $\theta=0$ is an alternating periodic chain model whose unit cell contains two sublattices called $A$ and $B$ sites. However, this is different from a dimerized chain because the hopping parameter is uniform throughout the chain. Please note that we have two different on-site energies instead of hopping parameters, unlike the SSH model.

Author response 6 We really appreciate the referee for giving detailed review of our manuscript. We follow the "Requested changes" and improve our manuscript. We summarize our changes following the "Requested changes'' in the section of ``List of important changes'' along with possible short comments.

List of important changes

1) We remove emphasis of the proposed experiments and add the references of the optical isolators. (Abstract) "$\cdots$ experimental realization of controllable localized, critical and delocalized states, using photonic crystals." $\to$ "[$\cdots$]. In addition, we suggest an experimental realization of controllable localized, critical and delocalized states, using photonic crystals.

(Caption of Fig.1) "Experimental control of localization characteristics in non-Hermitian optical system with ring resonators." $\to$ "A proposal for an experimental control of localization characteristics in non-Hermitian optical system with ring resonators."

"[$\cdots$] critical states are emphasized by [$\cdots$]" $\to$ "[$\cdots$] critical states are drawn by [$\cdots$]"

(Page 3, second paragraph) "Our main result is illustrated in Fig.1, with potential experimental implications." $\to$ "Fig. 1 illustrates our main results with sketch of potential experimental implications. "

"[$\cdots$] optical isolators [$\cdots$]'' $\to$ "[$\cdots$] optical isolators[``Encyclopedia of Physical Science and Technology'', ISBN: 978-0-12-227410-7][$\cdots$] "

(Page 15, second paragraph) "Importantly, our theoretical study could be studied by the photonic crystal\cite{PhysRevB.104.125416} or electrical circuits similar to other open system models governed by the Lindblad master equation or the effective non-Hermitian Hamiltonian. In particular, we propose an experimental setup to demonstrate the control of the localization of the wave function in the quasiperiodic system [$\cdots$]" $\to$ "Our theoretical work could be studied by the photonic crystal\cite{PhysRevB.104.125416} or electrical circuits similar to other open system models governed by the Lindblad master equation or the effective non-Hermitian Hamiltonian. In particular, we suggest an experimental setup to demonstrate the control of the localization of the wave function in the quasiperiodic system [$\cdots$]"

2) We collect all relevant definitions of phase rigidity, inverse participation ratios in section 2. We remove their detailed definitions in the section 3.

(Added---Page 4, Section 2) "One of the most important quantity used in the non-Hermitian systems is the phase rigidity, which is defined by

$$ r(\psi_k)=\vert\langle\psi_k^{(L)}|\psi_k^{(R)}\rangle\vert. $$
Here, the superscripts $L$ and $R$ stand for left and right eigenstates of the non-Hermitian Hamiltonian, and the subscript $k$ is the index of eigenstate. Unlike the Hermitian systems where the phase rigidity is always 1, it could be less than one, and even vanished in the non-Hermitian systems. Particularly, when two distinct eigenstates coalesce, the phase rigidity becomes zero\cite{PhysRevX.6.021007,PhysRevA.95.022117}. This unique characteristics of the non-Hermitian system is called an exceptional point. Thus, one can use the phase rigidity to quantify the coalesence of the states in the non-Hermitian system.

We quantify the localization strength of the state by using the inverse participation ratio (IPR), which is defined for a normalized state $\psi$ as

$$\mbox{IPR}(\psi)=\sum_i|\psi(i)|^4.$$
Note that the amount of localization for the wave function, $\psi$, can be quantified by the IPR\cite{PhysRevB.83.184206,calixto2015inverse,PhysRevB.100.054301}. In the spectrum, the maximum (minimum) value of the IPR indicates the maximally (minimally) localized state. Let us refer to these states in the spectrum as maximally localized and maximally extended states, respectively. Also, the average localization strength for entire states in the spectrum is given by the mean IPR (MIPR), defined by
$$\mbox{MIPR}=\frac{1}{N}\sum_{k=1}^{N}\mbox{IPR}(\psi_k),$$
where $\psi_k$ is the $k$-th eigenstate. The delocalization (localization) can be captured by the reduction (enhancement) of MIPR\cite{PhysRevB.100.054301}."

(Page 6, third paragraph) "Note that when $T=\frac{\Delta V}{4\cos(Ka/2)}$, $v_+(K)$ coalesces into $v_-(K)$. This coalescence is a unique feature of the non-Hermitian system, so-called an exceptional point (EP). When the states coalesce, the right and left eigenstates become orthogonal to each other, and hence the phase rigidity, $r(k)=\vert\langle v_+(k,L)|v_+(k,R)\rangle\vert$ becomes zero [PhysRevX.6.021007,PhysRevA.95.022117]. Here the subscripts $L$ and $R$ stand for the left and right eigenstates. Thus, the phase rigidity can be used to indicate the delocalization phase transition where the probability distribution becomes perfectly uniform." $\to$ "Note that when $T=\frac{\Delta V}{4\cos(Ka/2)}$, $v_+(K)$ coalesces into $v_-(K)$, and hence the phase rigidity defined in Eq. (2) for $v_\pm(K)$ becomes zero. Thus, the unconventional coalesence of the states due to the non-Hermiticity is happened when the state is uniformly delocalized. It turns out that the vanishment of the phase rigidity indicates the delocalization transition."

(Page 13, third paragraph) "[$\cdots$] To capture this, we compute the phase rigidity of the maximally localized state, given by $r(\Psi) =\left\vert\langle\Psi_L\vert\Psi_R\rangle\right\vert$, where $\Psi$ is the maximally localized state\cite{PhysRevX.6.021007,PhysRevA.95.022117}. The subscripts $L$ and $R$ denote the left and right eigenstates, respectively. Note that $r(\Psi)=1$ for the Hermitian case, while $r(\Psi)\le 1$ for the non-Hermitian case, because the right eigenstates could be non-orthogonal to each other. At the EPs where the multiple right eigenstates coalesce, the phase rigidity vanishes\cite{PhysRevX.6.021007,doi:10.1126/science.aar7709}." $\to$"[$\cdots$] To capture this, we compute the phase rigidity, $r(\Psi)$ of the maximally localized state. Remind that $r(\Psi)=1$ for the Hermitian case, while $r(\Psi)\le 1$ for the non-Hermitian case, because the right eigenstates could be non-orthonormal to each other."

(Removed---Page 7 in the previous version) "Specifically, we quantify the localization strength using the inverse participation ratio (IPR), which is defined for a normalized state $\psi$ as

$$ \mbox{IPR}(\psi)=\sum_i|\psi(i)|^4. $$
Note that the amount of localization for the wave function, $\psi$, can be quantified by the IPR\cite{PhysRevB.83.184206,calixto2015inverse,PhysRevB.100.054301}. In the spectrum, the maximum (minimum) value of the IPR indicates the maximally (minimally) localized state. Let us refer to these states in the spectrum as maximally localized and maximally extended states, respectively. Also, the average localization strength for entire states in the spectrum is given by the mean IPR (MIPR), defined by
$$\mbox{MIPR}=\frac{1}{N}\sum_{k=1}^{N}\mbox{IPR}(\psi_k),$$
where $\psi_k$ is the $k$-th eigenstate. The delocalization (localization) can be captured by the reduction (enhancement) of MIPR\cite{PhysRevB.100.054301}."

3) We move part of appendices into the main text. In addition, we clarify the captions of the figures.

(Page 13, last second paragraph) "[$\cdots$] For a given finite $T\ge 0.2V$, we generally see that the MIPR decreases as $\theta$ gets closer to $\pi/2$ in the Fibonacci quasicrystal. Thus, non-Hermiticity leads to the delocalization of states (see Appendix C.). [$\cdots$]"%Note that in the Fibonacci quasicrystal, even for the small $T$ regime, most of the states are critical states, which are delocalized as a power-law scaling. Thus, the hybridization of the eigenstates of the Hermitian Hamiltonian due to the non-reciprocal hopping phase occurs mainly between pairs of critical states in a way to compensate the power-law decaying probability amplitudes. This gives rise to the general delocalization tendency in terms of the non-reciprocal hopping phase." $\to$"[$\cdots$] For a given finite $T\ge 0.2V$, we generally see that the MIPR decreases as $\theta$ gets closer to $\pi/2$ in the Fibonacci quasicrystal (See Fig. 6). Thus, non-Hermiticity leads to the delocalization of states. In detail, Fig. 6 shows the MIPR as a function of the phase of the hopping parameter, $\theta$, for different hopping magnitudes $T$ in the Fibonacci quasicrystal. For the general $T$, the MIPR decreases as $\theta$ gets closer to $\pi/2$. Thus, the localization strength in the spectrum is suppressed in the Fibonacci quasicrystal due to the non-Hermiticity. [$\cdots$]"

(Caption of Fig. 6) "The mean value of the IPR (MIPR) of the spectrum as the function of the strength of the non-Hermiticity given by the phase angle of the hopping parameter, $\theta$. At $\theta=\pi/2$, the non-Hermiticity becomes maximum for given $T$. The MIPR which is the amount of the localization in the spectrum decreases as the non-Hermiticity becomes stronger." $\to$ "The mean value of the IPR (MIPR) of the energy spectrum as the function of the strength of the non-Hermiticity given by the phase angle of the hopping parameter, $\theta$ in the Fibonacci quasicrystal model. At $\theta=\pi/2$, the non-Hermiticity becomes maximum for given $T/V$, where $T$ is the hopping magnitude and $V=(V_A-V_B)/2$ is the difference between two kinds of the on-site energies, $V_A$ and $V_B$ in the Fibonacci quasicrystal model. Here, $V_A=1$ and $V_B=-1$. The MIPR (Eq. (4)) which is the amount of the localization in the spectrum decreases as the non-Hermiticity becomes stronger."

(Caption of Fig.7) "[$\cdots$] The degree of disorder is $50\%$. The system size, $N=233$, and the hopping parameter value, $T=4V$." $\to$ "[$\cdots$] The degree of disorder of the on-site potential energy is $50\%$. The system size, $N=233$, and the hopping parameter value, $T=4V=4$."

(Caption of Fig. 8) "[$\cdots$] as a function of the non-reciprocal hopping phase ($\theta$). Here, the unit of the localization length is the atomic spacing between neighboring atoms. The non-Hermiticity induces the delocalization, so the localization length increases as the non-Hermiticity becomes stronger. (b) Comparison of the probability distribution of the maximally localized states for (blue) $\theta=\pi/2$ and (red) $\theta=0$ in the logarithmic scale. The linear scaling in the figure indicates the exponential decay. The smaller slope indicates the larger localization length for the non-Hermitian case. The hopping magnitude is $T=3V$. The system size is $N=987$." $\to$ "[$\cdots$] as a function of the non-reciprocal hopping phase ($\theta$) in the Fibonacci quasicrystal model. Here, the unit of the localization length is the atomic spacing between neighboring atoms, which is set to be 1. The non-Hermiticity induces the delocalization, so the localization length increases as the non-Hermiticity becomes stronger. (b) Comparison of the probability distribution of the maximally localized states for (blue) $\theta=\pi/2$ and (red) $\theta=0$ in the logarithmic scale. The linear scaling in the figure indicates the exponential decay. The smaller slope indicates the larger localization length for the non-Hermitian case. The hopping magnitude is $T=3V=3$. The system size is $N=987$."

4) We remove the "return probability" Eq. (2) and related text. Instead, we explain the significance of the exceptional point at the beginning of section 3.

(Removed---Page 4, second paragraph of section 3 in the previous version) "More specifically, we denote $A_m$ as the transition amplitudes of the path traveling $m$ steps. In the discretized system, one can group the possible paths depending on the number of steps. Then the return probability for the $i$-th site, $P_i$, is given by

$$P_i=\left\vert\sum_{m=1}^{\infty}A_m\right\vert^2$$
Due to the non-reciprocal hopping phase, the phase difference between the transition amplitudes arises. Specifically, for $A_m$, the relative phase shift, $m\theta$ can be accumulated from $H_T$. In this case, the return probability of the state becomes smaller compared to the Hermitian case, indicating the delocalization of the state from the $i$-th site. On the other hand, the non-reciprocal phase can also increase the return probability due to constructive interference with respect to $\theta$ for some states. In this case, the delocalization is hindered by the interference from the non-reciprocal hopping phase. Thus, the non-reciprocal hopping phase gives rise to the \textit{state-dependent} control of the localization properties"

(Added---Page 5, first paragraph) "[$\cdots$] This interference originated from the non-reciprocal hopping phase essentially gives rise to the delocalization of the state by reducing the return probability. On the other hand, the non-reciprocal phase can also lead to the constructive interference with respect to $\theta$ for some states. In this case, the delocalization is hindered by the interference from the non-reciprocal hopping phase. Thus, the non-reciprocal hopping phase gives rise to the \textit{state-dependent} control of the localization properties. Moreover, in the non-Hermitian systems, such interference effect leads to the unconventional coalesence of the states which have different localization characteristics, so-called exceptional points. Hence, before and after this exceptional point, the localization characteristics would be changed drastically as we will demonstrate."

5) We do not use the unnecessary abbreviations such as "NHSE" and "EP".

6) We place Fig. 5 on the right place. We clarify the definition of $V$ and the relation to the AAF potential function in both caption and the main text.

(Page 10, last paragraph) "[$\cdots$] In detail, the Fibonacci quasicrystal consists of two different atoms, $A$ and $B$, which have onsite potentials $V_A$ and $V_B$, respectively. " $\to$"[$\cdots$] In detail, the Fibonacci quasicrystal consists of two different atoms, $A$ and $B$, which have onsite potentials $V_A$ and $V_B$, respectively. From Eq. (8), $V_A=\lambda$ and $V_B=-\lambda$. "

(Page 12, third paragraph) "The potential difference between atoms $A$ and $B$ is given by $V_A-V_B=2V$. " $\to$ "The potential difference between atoms $A$ and $B$ is given by $V_A-V_B=2V$. From Eq. (8), $V=\lambda$."

(Caption of Fig. 5) "[$\cdots$] with $T=13V$. The fractal dimensions are (b) $D_2=0$, (c) $D_2=0.411$ and (d) $D_2=1$. [$\cdots$] Along the boundary where the fractal dimension changes rapidly in Fig. (a), the phase rigidity also drops steeply in Fig. (e). [$\cdots$]" $\to$ "[$\cdots$] with $T=13V$. $V=(V_A-V_B)/2$ is the difference between two kinds of the on-site energies, $V_A$ and $V_B$ in the Fibonacci quasicrystal model. Here, $V_A=1$ and $V_B=-1$. The fractal dimensions are (b) $D_2=0$, (c) $D_2=0.411$ and (d) $D_2=1$. [$\cdots$] Along the boundary where the fractal dimension changes rapidly in panel (a), the phase rigidity also drops steeply in panel (e). [$\cdots$]"

7) We correct the list of references. We add missing DOIs and add ISBN for the books to clarify the types of the references.

8) We check the grammer and typos carefully, and correct them.

---

## Round 2 · Referee Report · Anonymous (Referee 3) · 2023-8-10

Report

I would very much appreciate the response and the corresponding revision of the manuscript. The authors have addressed all of my concerns satisfactorily and revised the manuscript accordingly. I would like to recommend publication of this manuscript in SciPost Physics.

  • validity: -
  • significance: -
  • originality: -
  • clarity: -
  • formatting: -
  • grammar: -

Author:  SungBin Lee  on 2023-09-05  [id 3953]

(in reply to Report 1 on 2023-08-10)

We appreciate the referee for giving a positive comment on our present manuscript.

---

## Round 2 · Referee Report · Anonymous (Referee 4) · 2023-8-23

Report

I would like to thank the authors for improving their presentation. These improvements have allowed me to better appreciate their work. Given that the issue of relevance is still under discussion, I have read the manuscript once again, and I am afraid that I have identified some further issues.

First, I admit that I confused dimerized hopping amplitudes and alternating on-site potentials in my previous Report, but I still believe that both are textbook physics in the Hermitian case.

In section 3.1, I really had some difficulty following the discussion when rereading it. For example, I was not able to verify statements such as "the delocalization of the state is generally observed" at the top of page 9 or "In particular, Fig.4(c) shows that the extended states disappear in the spectrum due to the non-reciprocal hopping phase" further down on the same page. Indeed, the IPR never goes to zero, i.e., the trends are only qualitative, but there is no strict transition and thus no strict "disappearance". Maybe this is due to the finite system size (finite approximant), and I believe that in order to establish a rigorous conclusion, the dependence on this parameter would have to be investigated. However, I have not even seen this parameter specified in section 3.1.

Actually, the biggest difference between the cases $\lambda=0.2$ and $2$ in Figs. 3(a) and 4(a) is the overall scale of the MIPR, but this difference is hidden by using different scales for the two cases. In fact, the biggest difference that I can see between Figs. 3 and 4 is the different shape of the $\lambda = 0.2$ curve in panel (a), but this is in fact a minor effect once the overall scales are considered. I also note that the color scales of panels (c) and (d) might suggest the opposite of what is actually shown (bright colors for small values, lighter ones for large values).

In section 3.2, the authors use a different approach from section 3.1 to characterize the $\beta\to \infty$ limit. In fact, reading statements such as "Although a larger IPR indicates stronger localization, the value of the IPR alone is insufficient to determine the detailed localization characteristics and scaling behavior of the wave function, since the IPR is an averaged quantity over space", I could not help wondering why they used exactly the criticized quantity in section 3.1. I think the relation between the approaches needs to be clarified, including the relation between Figs. 3 and 4 on the one side and Fig. 5 on the other side.

In section 3.2, the authors also include an analysis of the fractal dimension $D_2$. They explain that this quantity is derived from the scaling behavior with $N$, but again they do not specify the values of $N$ that they have used. I also suspect that working with finite $N$ gives rise to numerical errors for $D_2$. If so, these would have to be specified.

A final, more stylistic remark is that the authors write "we check the grammer and typos carefully, and correct them", I still have some issues with these (actually, this very sentence contains a spelling error). I would prefer to avoid providing a complete list of corrections and thus strongly recommend that they ask a colleague. If they cannot find a native English speaker, e.g., someone from Western Europe should already feel more comfortable with articles.

Requested changes

1- Specify size of the system/approximant in chapter 3.1 and discuss the influence of this parameter.
2- Clearly distinguish true transitions and crossovers in the discussion.
3- Explain the relevant features in Figs. 3 and 4 better, including differences and similarities.
4- If the authors do not wish to apply the same careful analysis as for the Fibonacci quasicrystal (section 3.2) also to the Aubry-André-Fibonacci model (section 3.1), they should at least explain better why they changed their mind when going to the $\beta\to\infty$ limit.
5- The values of $N$ used by the authors need to be specified in section 3.2; the resulting error of $D_2$ also needs to be discussed.
6- More on the level of detail, the "for the first time" on line 6 of the abstract is an unnecessary claim for priority. Consequently, I recommend removing it.
7- The English would still need proofreading. I will not provide a complete list of corrections here, but just some examples:
a) In the Abstract alone, I see at least three corrections:
i) Line 11: "two different limits" (add plural "s").
ii) Line 14: "the inverse participation ratio" (article "the" missing).
iii) Line 18: "and (2)" ("and" instead of comma - now these are just two items).
b) To the best of my knowledge "is happened" is not a correct English term (use "is happening" instead) and "vanishment" is not even an English word (maybe use "vanishing" instead).
I do recommend that the authors get some help with proofreading their article (the misspelled "controlloed" at the beginning of Appendix B should also be detected by a spellchecker).

  • validity: high
  • significance: good
  • originality: good
  • clarity: good
  • formatting: excellent
  • grammar: good

Author:  SungBin Lee  on 2023-09-05  [id 3952]

(in reply to Report 2 on 2023-08-23)
Category:
answer to question
correction

We attach the detailed author responses along with the list of changes as a PDF file.

*Author response 1
We thank the referee for detailed review. First of all, ``the delocalization of the state is generally observed'' is supported by the decrease of MIPR for both $\beta=0$ and $\beta=2.5$ with $T=2\lambda$ (See the red curves in Fig.3 (a) and Fig.4 (a)). The decrease of MIPR indicates that the delocalization of the states in the spectrum has been enhanced.

Next, "Fig.4(c)'' should be "Fig.4 (d)'' which shows the case of the maximally extended states. Note that if the maximally extended state is the localized state, then there are no extended states. It is because the maximally extended state has minimum value of the IPR. In particular, the sky-colored region in Fig.4 (d) corresponds to the exponentially localized states as we have already specified in the caption of Fig.4. To improve the clarity, we should also emphasize this fact in the main text, and demonstrate the localization of maximally extended state in the sky blue region in Fig.4 (d) in the appendix. The referee suspects that there is no strict disappearance of the extended states, however by investigating the localization properties of the maximally extended state in terms of the minimum value of the IPR in the spectrum, we can numerically specify the conditions where every eigenstate is localized.

The referee also asks the dependence on the system size, $N$. Although we specified $N=987$ for Figs.3 and 4, we should emphasize that the characteristics of the change of IPR as the function of $N$ does not change due to the non-reciprocal hopping phase in the AAF model. In other words, the fractal dimensions of the maximally localized and extended states do not change, except the case of the maximally extended states with $T/\lambda<0.3$ for $\beta=2.5$. Hence, the change of the system size only changes the whole values of the IPR, that is irrelevant to our main interests. Even for the case of the maximally extended states with $T/\lambda<0.3$ for $\beta=2.5$, the localization characteristics change from extended to exponentially localized without exploring the critical state. Since the IPR of the exponentially localized state is independent of $N$, the changing $N$ only decrease IPR of the extended states. This remains the localization and delocalization behavior as the function of $\theta$ shown in Fig.4 (d). Thus, the dependence on the system size, $N$ is irrelvant to our main interests in section 3.1.

Based on the referee's commet, we agree that the further clarifications and explanations are necessary. We modify our manuscript to improve the clarity. For clarification, we emphasize that the sky blue region in Fig. 4 (d) represents the localized state not only in the caption but also in the main text. We also clarify the dependence on the system size $N$, and its irrelevance to our main interests.

*Author response 2
First of all, the different scales between $T=0.2\lambda$ and $2\lambda$ are not significant because depending on the $T/\lambda$, the phases are different. For example, as we have explained in the last second paragraph on page 7, for $T/\lambda<0.5$ and $\beta=0$, every state is localized for the Hermitian case. On the other hand, for $T/\lambda>0.5$, every state is extended. These facts for the Hermitian case are already known, thus, the discussion of the scales of the MIPR for different $T$ is not necessary here.

As the referee has pointed out, one of the notable difference between Figs. 3 and 4 is the different shape of the $T=0.2\lambda$ curve in panel (a). The referee claims that this is a minor effect once the overall scales, however, this is not a minor difference. Although the fraction of the change of MIPR is small, the different types of the change i.e. decreasing and increasing gives totally different results for each state. Specifically, the delocalization of the maximally extended state is observed for $\beta=0$, while the exponential localization tendency of maximally extended states with varying $\theta$ occurs for $\beta=2.5$ case i.e. localization. Note that the number of extended states, which become localized near $\theta=\pi/2$ for $\beta=2.5$ is small for the Hermitian case when $T/\lambda$ is small. Thus, even for the case of $\beta=2.5$, the fraction of change on MIPR is not large. However, this never means that the different types of the changes on the MIPR for $\beta=0$ and $\beta=2.5$ with $T=0.2\lambda$ are not important. We emphasize that the different types of the changes on the MIPR capture the totally different localization tendancies of each state as the functions of $\theta$.

*Author response 3
We appreciate your comment. Indeed, we think it is better to have more explanation for clarification in the main text. We have not used the fractal dimension in section 3.1 because we do not have any critical states and phase transitions exploring unconventional fractal dimensions. Please note that the fractal dimension is important when there are some fractal wave functions such as the critical states. However, in the AAF model discussed in section 3.1 have either localized or extended states only. Furthermore, although the strength of the localization given by the IPR would be changed, the characteristics of the localization, which are termed by extended, localized and critical are not altered by $\theta$ in almost all of the cases. For instance, note that the maximally localized states, which are exponentially localized for $\theta=0$ remain localized as we have mentioned in the main text. The only case where the fractal dimension changes as $\theta$ approaches $\pi/2$ is the maximally extended states for small $T/\lambda$ regime with $\beta=2.5$ shown in Fig. 4 (d). However, again even in this case, we have only two trivial kinds of the localization characteristics, localized and extended whose localization characteristics do not have the fractality. Thus, we believe that it is enough to explicitly show the localized and extended states, respectively in the appendix. The landscape of the fractal dimension does not give any further information to the readers in this case. Instead, for clarity, we specify the localization characteristics of the states either exponentially localized ($D_2=0$) or extended ($D_2=1$) in the captions of Figs 3 and 4.

On the other hand, for the case of the Fibonacci tiling, we have found the important delocalization transition exploring unconventional fractal dimension as shown in Fig. 5. Furthermore, in the case of $\beta\to\infty$, the maximally localized state would be not only localized or extended but also critical states having intermediate fractal dimensions $D_2\approx 0.5$. Thus, the discussion using the fractal dimension is relevant to the case of Fibonacci tiling only. We agree that it is better to explain the importance of different approaches as the referee commented. We briefly specify the significance of the fractal dimension when we discuss the Fibonacci quasicrystal in the main text.

*Author response 4
Based on the referee's comment, we add the information of $N$ values we have used to compute $D_2$. Surely, the fractal dimension has some numerical errors with finite $N$. However, such error does not change our results shown in Fig. 5. We also add the detailed numerical results relevant to Fig. 5 in the appendix.

*Author response 5
We have asked one of our colleague who is native in English. Based on his comment, we corrected typos and expressions.

Attachment:

Author_response_scipost.pdf

---

## Round 2 · Author Response

We appreciate the referee for giving a positive comment on our present manuscript.

Author response 1.

We thanks for the detailed review of our manuscript. Let us briefly emphasize the novelty and importance of our work. Recently, the non-Hermitian physics is widely studied in physics of the open systems such as optics and condensed matter physics. One of the major stream of the research in the non-Hermitian physics is the interplay of disorder and non-Hermiticity. However, the non-Hermitian system with non-reciprocal hopping phases has never been explored, and its interplay with the disorder or quasiperiodic order has been elusive. In this context, our findings of the phase transition from localized phase to delocalized phase due to the non-reciprocal hopping phases on the quasiperiodic system can give an important contribution in the general research areas related to the non-Hermitian physics and open systems. Hence, we believe that our work is suitable for SciPost Physics.

Author response 2.
We agree with the comment on our Fig. 1 and abstract raised by the second referee. Thus, we edited our manuscript in the section where we discuss the possible experimental applications and noted that Fig.1 is the potential experimental application.

The second referee asked if it is really possible to propagate light only in one direction. This is possible by using the optical isolator (or sometimes called optical diode), which is generally used to study the non-reciprocal optics. An optical isolator is a device that allows light to propagate through it only in one direction, but not in the opposite direction. Isolators are useful as valves that allow propagation in only one direction. They are used in high-power applications, for which one desires one-way transmission of light. [Refer to the book, ``Encyclopedia of Physical Science and Technology'', ISBN: 978-0-12-227410-7]. We have added this reference, in the section we introduce the optical isolator. Hence, we believe that our suggestion of the possible experiment realization shown in Fig. 1 would be reasonable, and we are currently in communication with experimentalists in this field.

Author response 3
Based on the second referee's comments, we have edited our manuscript for better understanding. Specifically, we expand the section 2 with including the basic definitions of the phase rigidity and inverse participation ratio (IPR). We also removed the redundant expressions in our manuscript. We also move the appendix C to the main text for better understanding. The relevant quantities such as mean inverse participation ratio (MIPR) are also introduced in section 2. We leave the appendix A as the appendix because it would interrupt the flow of the main text. We specify the model (random disordered chain or the Fibonacci chain), and added detailed numerical values in the captions of the figures.

Author response 4
As the second referee has pointed out, the definition of the ``return probability'' in Eq.(2) has not been directly used except specifying the meaning of the interference due to the non-reciprocal hopping phases. Thus, based on the referee's comment, we remove the definition (Eq. (2)) and the related detailed discussion. Instead, we briefly explain how the coalesence of the states would change the localization characteristics of the states, which is one of the important notions that appear to explain our results.

Author response 5
The model Eq. (3) with $\theta=0$ is an alternating periodic chain model whose unit cell contains two sublattices called $A$ and $B$ sites. However, this is different from a dimerized chain because the hopping parameter is uniform throughout the chain. Please note that we have two different on-site energies instead of hopping parameters, unlike the SSH model.

Author response 6
The terminology of the “phase rigidity” is widely used in the non-Hermitian physics. This measures the rigidity of the phase of eigenfunction. In the non-Hermitian system, the eigenfunctions are biorthonormal i.e.
$$\langle\psi_n^L|\psi_m^R\rangle=\delta_{nm}, $$ where $$\vert\psi^R\rangle=\frac{\vert\phi^R\rangle}{\sqrt{\langle\phi^L|\phi^R\rangle}} \ \mbox{and} \vert\psi^L\rangle=\frac{\vert\phi^L\rangle}{\sqrt{\langle\phi^L|\phi^R\rangle}} $$
Here, $R$ and $L$ stand for the right and left eigenstates, $H\vert\phi^R\rangle=E\vert\phi^R\rangle$ and $\langle\phi^L\vert H=E\langle\phi^L\vert$, where $H$ is the non-Hermitian Hamiltonian. $n,m$ are indices of the eigenstates.
However, at the exceptional point (zero phase rigidity), where two states coalescence, the states become linearly dependent, and the biorthonormal condition is failed. At this point, their phase (in a sense of the complex number) jumps---which is known as the characteristics of the branch point. Due to this jump of the phase, the phase of the eigenfunction is not rigid at vicinity of the exceptional points. Thus, we call it as the phase rigidity. Further details are explained in the review paper, [Ingrid Rotter 2009 J. Phys. A: Math. Theor. 42 153001].

Author response 7
We appreciate for pointing out the typos and grammar mistakes. We check our manuscript carefully and correct such typos and grammatical errors.

---

## Round 2 · List of Changes

1) We remove emphasis of the proposed experiments and add the references of the optical isolators.
(Abstract)
"[$\cdots$](3) experimental realization of controllable localized, critical and delocalized states, using photonic crystals."
$\to$ "[$\cdots$]. In addition, we suggest an experimental realization of controllable localized, critical and delocalized states, using photonic crystals.

(Caption of Fig.1)
"Experimental control of localization characteristics in non-Hermitian optical system with ring resonators."
$\to$ "A proposal for an experimental control of localization characteristics in non-Hermitian optical system with ring resonators."

"[$\cdots$] critical states are emphasized by [$\cdots$]"
$\to$ "[$\cdots$] critical states are drawn by [$\cdots$]"

(Page 3, second paragraph)
"Our main result is illustrated in Fig.1, with potential experimental implications."
$\to$ "Fig. 1 illustrates our main results with sketch of potential experimental implications. "

"[$\cdots$] optical isolators [$\cdots$]''
$\to$ "[$\cdots$] optical isolators[``Encyclopedia of Physical Science and Technology'', ISBN: 978-0-12-227410-7][$\cdots$] "

(Page 15, second paragraph)
"Importantly, our theoretical study could be studied by the photonic crystal\cite{PhysRevB.104.125416} or electrical circuits similar to other open system models governed by the Lindblad master equation or the effective non-Hermitian Hamiltonian. In particular, we propose an experimental setup to demonstrate the control of the localization of the wave function in the quasiperiodic system [$\cdots$]"
$\to$ "Our theoretical work could be studied by the photonic crystal\cite{PhysRevB.104.125416} or electrical circuits similar to other open system models governed by the Lindblad master equation or the effective non-Hermitian Hamiltonian. In particular, we suggest an experimental setup to demonstrate the control of the localization of the wave function in the quasiperiodic system [$\cdots$]"

2) We collect all relevant definitions of phase rigidity, inverse participation ratios in section 2. We remove their detailed definitions in the section 3.

(Added---Page 4, Section 2)
"One of the most important quantity used in the non-Hermitian systems is the phase rigidity, which is defined by
$$
r(\psi_k)=\vert\langle\psi_k^{(L)}|\psi_k^{(R)}\rangle\vert.
$$
Here, the superscripts $L$ and $R$ stand for left and right eigenstates of the non-Hermitian Hamiltonian, and the subscript $k$ is the index of eigenstate. Unlike the Hermitian systems where the phase rigidity is always 1, it could be less than one, and even vanished in the non-Hermitian systems. Particularly, when two distinct eigenstates coalesce, the phase rigidity becomes zero\cite{PhysRevX.6.021007,PhysRevA.95.022117}. This unique characteristics of the non-Hermitian system is called an exceptional point. Thus, one can use the phase rigidity to quantify the coalesence of the states in the non-Hermitian system.

We quantify the localization strength of the state by using the inverse participation ratio (IPR), which is defined for a normalized state $\psi$ as
$$\mbox{IPR}(\psi)=\sum_i|\psi(i)|^4.$$
Note that the amount of localization for the wave function, $\psi$, can be quantified by the IPR\cite{PhysRevB.83.184206,calixto2015inverse,PhysRevB.100.054301}. In the spectrum, the maximum (minimum) value of the IPR indicates the maximally (minimally) localized state. Let us refer to these states in the spectrum as maximally localized and maximally extended states, respectively. Also, the average localization strength for entire states in the spectrum is given by the mean IPR (MIPR), defined by
$$\mbox{MIPR}=\frac{1}{N}\sum_{k=1}^{N}\mbox{IPR}(\psi_k),$$
where $\psi_k$ is the $k$-th eigenstate. The delocalization (localization) can be captured by the reduction (enhancement) of MIPR\cite{PhysRevB.100.054301}."

(Page 6, third paragraph)
"Note that when $T=\frac{\Delta V}{4\cos(Ka/2)}$, $v_+(K)$ coalesces into $v_-(K)$. This coalescence is a unique feature of the non-Hermitian system, so-called an exceptional point (EP). When the states coalesce, the right and left eigenstates become orthogonal to each other, and hence the phase rigidity, $r(k)=\vert\langle v_+(k,L)|v_+(k,R)\rangle\vert$ becomes zero [PhysRevX.6.021007,PhysRevA.95.022117]. Here the subscripts $L$ and $R$ stand for the left and right eigenstates. Thus, the phase rigidity can be used to indicate the delocalization phase transition where the probability distribution becomes perfectly uniform."
$\to$ "Note that when $T=\frac{\Delta V}{4\cos(Ka/2)}$, $v_+(K)$ coalesces into $v_-(K)$, and hence the phase rigidity defined in Eq. (2) for $v_\pm(K)$ becomes zero. Thus, the unconventional coalesence of the states due to the non-Hermiticity is happened when the state is uniformly delocalized. It turns out that the vanishment of the phase rigidity indicates the delocalization transition."

(Page 13, third paragraph)
"[$\cdots$] To capture this, we compute the phase rigidity of the maximally localized state, given by $r(\Psi) =\left\vert\langle\Psi_L\vert\Psi_R\rangle\right\vert$, where $\Psi$ is the maximally localized state\cite{PhysRevX.6.021007,PhysRevA.95.022117}. The subscripts $L$ and $R$ denote the left and right eigenstates, respectively. Note that $r(\Psi)=1$ for the Hermitian case, while $r(\Psi)\le 1$ for the non-Hermitian case, because the right eigenstates could be non-orthogonal to each other. At the EPs where the multiple right eigenstates coalesce, the phase rigidity vanishes\cite{PhysRevX.6.021007,doi:10.1126/science.aar7709}."
$\to$"[$\cdots$] To capture this, we compute the phase rigidity, $r(\Psi)$ of the maximally localized state. Remind that $r(\Psi)=1$ for the Hermitian case, while $r(\Psi)\le 1$ for the non-Hermitian case, because the right eigenstates could be non-orthonormal to each other."

(Removed---Page 7 in the previous version)
"Specifically, we quantify the localization strength using the inverse participation ratio (IPR), which is defined for a normalized state $\psi$ as
$$
\mbox{IPR}(\psi)=\sum_i|\psi(i)|^4.
$$
Note that the amount of localization for the wave function, $\psi$, can be quantified by the IPR\cite{PhysRevB.83.184206,calixto2015inverse,PhysRevB.100.054301}. In the spectrum, the maximum (minimum) value of the IPR indicates the maximally (minimally) localized state. Let us refer to these states in the spectrum as maximally localized and maximally extended states, respectively. Also, the average localization strength for entire states in the spectrum is given by the mean IPR (MIPR), defined by
$$\mbox{MIPR}=\frac{1}{N}\sum_{k=1}^{N}\mbox{IPR}(\psi_k),$$
where $\psi_k$ is the $k$-th eigenstate. The delocalization (localization) can be captured by the reduction (enhancement) of MIPR\cite{PhysRevB.100.054301}."

3) We move part of appendices into the main text. In addition, we clarify the captions of the figures.

(Page 13, last second paragraph)
"[$\cdots$] For a given finite $T\ge 0.2V$, we generally see that the MIPR decreases as $\theta$ gets closer to $\pi/2$ in the Fibonacci quasicrystal. Thus, non-Hermiticity leads to the delocalization of states (see Appendix C.). [$\cdots$]"%Note that in the Fibonacci quasicrystal, even for the small $T$ regime, most of the states are critical states, which are delocalized as a power-law scaling. Thus, the hybridization of the eigenstates of the Hermitian Hamiltonian due to the non-reciprocal hopping phase occurs mainly between pairs of critical states in a way to compensate the power-law decaying probability amplitudes. This gives rise to the general delocalization tendency in terms of the non-reciprocal hopping phase."
$\to$"[$\cdots$] For a given finite $T\ge 0.2V$, we generally see that the MIPR decreases as $\theta$ gets closer to $\pi/2$ in the Fibonacci quasicrystal (See Fig. 6). Thus, non-Hermiticity leads to the delocalization of states. In detail, Fig. 6 shows the MIPR as a function of the phase of the hopping parameter, $\theta$, for different hopping magnitudes $T$ in the Fibonacci quasicrystal. For the general $T$, the MIPR decreases as $\theta$ gets closer to $\pi/2$. Thus, the localization strength in the spectrum is suppressed in the Fibonacci quasicrystal due to the non-Hermiticity. [$\cdots$]"

(Caption of Fig. 6)
"The mean value of the IPR (MIPR) of the spectrum as the function of the strength of the non-Hermiticity given by the phase angle of the hopping parameter, $\theta$. At $\theta=\pi/2$, the non-Hermiticity becomes maximum for given $T$. The MIPR which is the amount of the localization in the spectrum decreases as the non-Hermiticity becomes stronger."
$\to$ "The mean value of the IPR (MIPR) of the energy spectrum as the function of the strength of the non-Hermiticity given by the phase angle of the hopping parameter, $\theta$ in the Fibonacci quasicrystal model. At $\theta=\pi/2$, the non-Hermiticity becomes maximum for given $T/V$, where $T$ is the hopping magnitude and $V=(V_A-V_B)/2$ is the difference between two kinds of the on-site energies, $V_A$ and $V_B$ in the Fibonacci quasicrystal model. Here, $V_A=1$ and $V_B=-1$. The MIPR (Eq. (4)) which is the amount of the localization in the spectrum decreases as the non-Hermiticity becomes stronger."

(Caption of Fig.7)
"[$\cdots$] The degree of disorder is $50\%$. The system size, $N=233$, and the hopping parameter value, $T=4V$."
$\to$ "[$\cdots$] The degree of disorder of the on-site potential energy is $50\%$. The system size, $N=233$, and the hopping parameter value, $T=4V=4$."

(Caption of Fig. 8)
"[$\cdots$] as a function of the non-reciprocal hopping phase ($\theta$). Here, the unit of the localization length is the atomic spacing between neighboring atoms. The non-Hermiticity induces the delocalization, so the localization length increases as the non-Hermiticity becomes stronger. (b) Comparison of the probability distribution of the maximally localized states for (blue) $\theta=\pi/2$ and (red) $\theta=0$ in the logarithmic scale. The linear scaling in the figure indicates the exponential decay. The smaller slope indicates the larger localization length for the non-Hermitian case. The hopping magnitude is $T=3V$. The system size is $N=987$."
$\to$ "[$\cdots$] as a function of the non-reciprocal hopping phase ($\theta$) in the Fibonacci quasicrystal model. Here, the unit of the localization length is the atomic spacing between neighboring atoms, which is set to be 1. The non-Hermiticity induces the delocalization, so the localization length increases as the non-Hermiticity becomes stronger. (b) Comparison of the probability distribution of the maximally localized states for (blue) $\theta=\pi/2$ and (red) $\theta=0$ in the logarithmic scale. The linear scaling in the figure indicates the exponential decay. The smaller slope indicates the larger localization length for the non-Hermitian case. The hopping magnitude is $T=3V=3$. The system size is $N=987$."

4) We remove the "return probability" Eq. (2) and related text. Instead, we explain the significance of the exceptional point at the beginning of section 3.

(Removed---Page 4, second paragraph of section 3 in the previous version)
"More specifically, we denote $A_m$ as the transition amplitudes of the path traveling $m$ steps. In the discretized system, one can group the possible paths depending on the number of steps. Then the return probability for the $i$-th site, $P_i$, is given by
$$P_i=\left\vert\sum_{m=1}^{\infty}A_m\right\vert^2$$
Due to the non-reciprocal hopping phase, the phase difference between the transition amplitudes arises. Specifically, for $A_m$, the relative phase shift, $m\theta$ can be accumulated from $H_T$. In this case, the return probability of the state becomes smaller compared to the Hermitian case, indicating the delocalization of the state from the $i$-th site. On the other hand, the non-reciprocal phase can also increase the return probability due to constructive interference with respect to $\theta$ for some states. In this case, the delocalization is hindered by the interference from the non-reciprocal hopping phase. Thus, the non-reciprocal hopping phase gives rise to the \textit{state-dependent} control of the localization properties"

(Added---Page 5, first paragraph)
"[$\cdots$] This interference originated from the non-reciprocal hopping phase essentially gives rise to the delocalization of the state by reducing the return probability. On the other hand, the non-reciprocal phase can also lead to the constructive interference with respect to $\theta$ for some states. In this case, the delocalization is hindered by the interference from the non-reciprocal hopping phase. Thus, the non-reciprocal hopping phase gives rise to the \textit{state-dependent} control of the localization properties. Moreover, in the non-Hermitian systems, such interference effect leads to the unconventional coalesence of the states which have different localization characteristics, so-called exceptional points. Hence, before and after this exceptional point, the localization characteristics would be changed drastically as we will demonstrate."

5) We do not use the unnecessary abbreviations such as "NHSE" and "EP".

6) We place Fig. 5 on the right place. We clarify the definition of $V$ and the relation to the AAF potential function in both caption and the main text.

(Page 10, last paragraph)
"[$\cdots$] In detail, the Fibonacci quasicrystal consists of two different atoms, $A$ and $B$, which have onsite potentials $V_A$ and $V_B$, respectively. "
$\to$"[$\cdots$] In detail, the Fibonacci quasicrystal consists of two different atoms, $A$ and $B$, which have onsite potentials $V_A$ and $V_B$, respectively. From Eq. (8), $V_A=\lambda$ and $V_B=-\lambda$. "

(Page 12, third paragraph)
"The potential difference between atoms $A$ and $B$ is given by $V_A-V_B=2V$. "
$\to$ "The potential difference between atoms $A$ and $B$ is given by $V_A-V_B=2V$. From Eq. (8), $V=\lambda$."

(Caption of Fig. 5)
"[$\cdots$] with $T=13V$. The fractal dimensions are (b) $D_2=0$, (c) $D_2=0.411$ and (d) $D_2=1$. [$\cdots$] Along the boundary where the fractal dimension changes rapidly in Fig. (a), the phase rigidity also drops steeply in Fig. (e). [$\cdots$]"
$\to$ "[$\cdots$] with $T=13V$. $V=(V_A-V_B)/2$ is the difference between two kinds of the on-site energies, $V_A$ and $V_B$ in the Fibonacci quasicrystal model. Here, $V_A=1$ and $V_B=-1$. The fractal dimensions are (b) $D_2=0$, (c) $D_2=0.411$ and (d) $D_2=1$. [$\cdots$] Along the boundary where the fractal dimension changes rapidly in panel (a), the phase rigidity also drops steeply in panel (e). [$\cdots$]"

7) We correct the list of references. We add missing DOIs and add ISBN for the books to clarify the types of the references.

8) We check the grammer and typos carefully, and correct them.

---

## Round 3 · Referee Report · Anonymous (Referee 2) · 2023-10-15

Report

The authors continue to improve their manuscript. Nevertheless, some issues still remain.

1- I believe that it gets increasingly clear that the results of section 3.1 are only qualitative: the data does not allow a strict distinction between localized and extended states, only the identification of qualitative trends. This is aggravated by statements being made that might be supported by the data, but are difficult to extract from the figures that they are attributed to. For example, I believe that it is difficult if not impossible to extract any trend as a function of $\theta$ from Figs. 3(c,d) and Fig. 4(c) scanning across, e.g., $\lambda=0.2$ since the information is simply not contained in the color scale. I admit that I might have made this observation before, but after each round of revision I am slowly understanding better what the authors really mean to say.

2- The discussion in section 3.2 is more quantitative. Still, the new data for the fractal dimension $D_2$ in appendix D shows that it is extracted from sizes $300 \le N \le 987$. This is only a factor just above 3 in system size while it is consensus in critical phenomena that several orders of magnitude are needed to firmly establish power laws. The conclusions may be correct, but the data presented in the manuscript does not suffice to exclude that the estimates for $D_2$ are affected by corrections to scaling. Furthermore, I believe that one should be able to push computations to $N >987$ without too much effort.

We could continue reviewing and revising the manuscript. However, I believe that review should be closed at this point. In my opinion, in view of the above reservations, the manuscript does not meet the standards of SciPost Physics, but it could be published in SciPost Physics Core in its present form.

Requested changes

Minor typographic issues to be corrected on the proofs: 1- Last line of second paragraph on page 2: "quasiperiodic" is misspelled. 2- First line of section 2: no hyphen in "$N$ sites". 3- Fifth line of third paragraph of section 3.2: no comma after "Eq. (8)", i.e., "Eq. (8).". 4- Third line of second paragraph on page 13: I believe that "Recall" would be more appropriate than "Remind". 5- If the production team does not fix punctuation, the authors might pay attention to eliminating full stops inside parentheses if there is one following it (at least two instances of ".).".

---

## Round 3 · Author Response

*Author response 1
We appreciate the first referee for giving a positive comment on our present manuscript.

*Author response 2
We thank the referee for detailed review. First of all, ``the delocalization of the state is generally observed'' is supported by the decrease of MIPR for both $\beta=0$ and $\beta=2.5$ with $T=2\lambda$ (See the red curves in Fig.3 (a) and Fig.4 (a)). The decrease of MIPR indicates that the delocalization of the states in the spectrum has been enhanced.

Next, "Fig.4(c)'' should be "Fig.4 (d)'' which shows the case of the maximally extended states. Note that if the maximally extended state is the localized state, then there are no extended states. It is because the maximally extended state has minimum value of the IPR. In particular, the sky-colored region in Fig.4 (d) corresponds to the exponentially localized states as we have already specified in the caption of Fig.4. To improve the clarity, we should also emphasize this fact in the main text, and demonstrate the localization of maximally extended state in the sky blue region in Fig.4 (d) in the appendix. The referee suspects that there is no strict disappearance of the extended states, however by investigating the localization properties of the maximally extended state in terms of the minimum value of the IPR in the spectrum, we can numerically specify the conditions where every eigenstate is localized.

The referee also asks the dependence on the system size, $N$. Although we specified $N=987$ for Figs.3 and 4, we should emphasize that the characteristics of the change of IPR as the function of $N$ does not change due to the non-reciprocal hopping phase in the AAF model. In other words, the fractal dimensions of the maximally localized and extended states do not change, except the case of the maximally extended states with $T/\lambda<0.3$ for $\beta=2.5$. Hence, the change of the system size only changes the whole values of the IPR, that is irrelevant to our main interests. Even for the case of the maximally extended states with $T/\lambda<0.3$ for $\beta=2.5$, the localization characteristics change from extended to exponentially localized without exploring the critical state. Since the IPR of the exponentially localized state is independent of $N$, the changing $N$ only decrease IPR of the extended states. This remains the localization and delocalization behavior as the function of $\theta$ shown in Fig.4 (d). Thus, the dependence on the system size, $N$ is irrelvant to our main interests in section 3.1.

Based on the referee's commet, we agree that the further clarifications and explanations are necessary. We modify our manuscript to improve the clarity. For clarification, we emphasize that the sky blue region in Fig. 4 (d) represents the localized state not only in the caption but also in the main text. We also clarify the dependence on the system size $N$, and its irrelevance to our main interests.

*Author response 3
First of all, the different scales between $T=0.2\lambda$ and $2\lambda$ are not significant because depending on the $T/\lambda$, the phases are different. For example, as we have explained in the last second paragraph on page 7, for $T/\lambda<0.5$ and $\beta=0$, every state is localized for the Hermitian case. On the other hand, for $T/\lambda>0.5$, every state is extended. These facts for the Hermitian case are already known, thus, the discussion of the scales of the MIPR for different $T$ is not necessary here.

As the referee has pointed out, one of the notable difference between Figs. 3 and 4 is the different shape of the $T=0.2\lambda$ curve in panel (a). The referee claims that this is a minor effect once the overall scales, however, this is not a minor difference. Although the fraction of the change of MIPR is small, the different types of the change i.e. decreasing and increasing gives totally different results for each state. Specifically, the delocalization of the maximally extended state is observed for $\beta=0$, while the exponential localization tendency of maximally extended states with varying $\theta$ occurs for $\beta=2.5$ case i.e. localization. Note that the number of extended states, which become localized near $\theta=\pi/2$ for $\beta=2.5$ is small for the Hermitian case when $T/\lambda$ is small. Thus, even for the case of $\beta=2.5$, the fraction of change on MIPR is not large. However, this never means that the different types of the changes on the MIPR for $\beta=0$ and $\beta=2.5$ with $T=0.2\lambda$ are not important. We emphasize that the different types of the changes on the MIPR capture the totally different localization tendancies of each state as the functions of $\theta$.

*Author response 4
We appreciate your comment. Indeed, we think it is better to have more explanation for clarification in the main text. We have not used the fractal dimension in section 3.1 because we do not have any critical states and phase transitions exploring unconventional fractal dimensions. Please note that the fractal dimension is important when there are some fractal wave functions such as the critical states. However, in the AAF model discussed in section 3.1 have either localized or extended states only. Furthermore, although the strength of the localization given by the IPR would be changed, the characteristics of the localization, which are termed by extended, localized and critical are not altered by $\theta$ in almost all of the cases. For instance, note that the maximally localized states, which are exponentially localized for $\theta=0$ remain localized as we have mentioned in the main text. The only case where the fractal dimension changes as $\theta$ approaches $\pi/2$ is the maximally extended states for small $T/\lambda$ regime with $\beta=2.5$ shown in Fig. 4 (d). However, again even in this case, we have only two trivial kinds of the localization characteristics, localized and extended whose localization characteristics do not have the fractality. Thus, we believe that it is enough to explicitly show the localized and extended states, respectively in the appendix. The landscape of the fractal dimension does not give any further information to the readers in this case. Instead, for clarity, we specify the localization characteristics of the states either exponentially localized ($D_2=0$) or extended ($D_2=1$) in the captions of Figs 3 and 4.

On the other hand, for the case of the Fibonacci tiling, we have found the important delocalization transition exploring unconventional fractal dimension as shown in Fig. 5. Furthermore, in the case of $\beta\to\infty$, the maximally localized state would be not only localized or extended but also critical states having intermediate fractal dimensions $D_2\approx 0.5$. Thus, the discussion using the fractal dimension is relevant to the case of Fibonacci tiling only. We agree that it is better to explain the importance of different approaches as the referee commented. We briefly specify the significance of the fractal dimension when we discuss the Fibonacci quasicrystal in the main text.

*Author response 5
We thank for pointing it out. Based on the referee's comment, we add the information of $N$ values we have used to compute $D_2$. Surely, the fractal dimension has some numerical errors with finite $N$. However, such error does not change our results shown in Fig. 5. We also add the detailed numerical results relevant to Fig. 5 in the appendix.

*Author response 6
We thank again for your comment. We have asked one of our colleague who is native in English. Based on his comment, we corrected typos and expressions.

---

## Round 3 · List of Changes

1. We specify the size of the system in chapter 3.1, and clarify its insignificance.

  2. We clarify the transition where the extended states disappear, supported by FIg.4 (d) in the main text.

  3. We explain the relevant features in Figs. 3 and 4 better, specifying difference and similarities.

  4. We briefly specify the reason why the discussion using the fractal dimension is relevant to the Fibonacci case ($\beta\to\infty$). Also, in the captions of Figs.3 and 4, we specify the localization characteristics of the states discussed in section 3.1, either exponentially localized or extended. In detail, every maximally localized states are exponentially localized regardless of $\beta$ and $T/\lambda$. For $T\ge 0.5\lambda$, every maximally extended states are extended. For $\beta=0$ and $T<0.5\lambda$, every maximally extended states are exponentially localized. On the other hand, for $\beta=2.5$ and $T<0.5\lambda$, the sky blue region indicates the exponentially localized states, while the green and black regions indicate the extended states.

  5. We specify the values of $N$ we used to compute the fractal dimensions. In the appendix, we add the figures to clarify the numerical error (which is minor) of the fractal dimensions. In addition, to satisfy the referee's concern, we also add the error bars on Fig. 7 for the random disordered chain. Note that this error bar is originated from the random disorders. We use 20 samples of random local potential distributions.

  6. We corrected the typos.

*A more detailed list of changes as a PDF file is attached as the author's response to the second reviewer's comment.

---

## Editorial Decision

published